# Boosting thermo-photocatalytic $CO_2$ conversion activity by using photosynthesis-inspired electron-proton-transfer mediators

Yingxuan Li [1✉], Danping Hui[1], Yuqing Sun[1], Ying Wang[2✉], Zhijian Wu[2], Chuanyi Wang[1] & Jincai Zhao[3]

Natural photosynthesis proceeded by sequential water splitting and $CO_2$ reduction reactions is an efficient strategy for $CO_2$ conversion. Here, mimicking photosynthesis to boost $CO_2$-to-CO conversion is achieved by using plasmonic Bi as an electron-proton-transfer mediator. Electroreduction of $H_2O$ with a Bi electrode simultaneously produces $O_2$ and hydrogen-stored Bi (Bi-$H_x$). The obtained Bi-$H_x$ is subsequently used to generate electron-proton pairs under light irradiation to reduce $CO_2$ to CO; meanwhile, Bi-$H_x$ recovers to Bi, completing the catalytic cycle. This two-step strategy avoids $O_2$ separation and enables a CO production efficiency of 283.8 µmol g$^{-1}$ h$^{-1}$ without sacrificial reagents and cocatalysts, which is 9 times that on pristine Bi in $H_2$ gas. Theoretical/experimental studies confirm that such excellent activity is attributed to the formed Bi-$H_x$ intermediate that improves charge separation and reduces reaction barriers in $CO_2$ reduction.

[1] School of Environmental Science and Engineering, Shaanxi University of Science and Technology, Xi'an 710021, China. [2] State Key Laboratory of Rare Earth Resource Utilization, Changchun Institute of Applied Chemistry, Chinese Academy of Sciences, Changchun 130022, China. [3] Key Laboratory of Photochemistry, CAS Research/Education Center for Excellence in Molecular Sciences, Institute of Chemistry, Chinese Academy of Sciences, Beijing 100190, China. ✉email: liyingxuan@sust.edu.cn; ywang_2012@ciac.ac.cn

atalytic $CO_2$ reduction driven by solar light or renewable electricity to produce useful fuels is a promising strategy for solving energy and greenhouse effect issues[1]. The $CO_2$ reduction activity is usually limited by the rate-limiting step of transferring an electron to a linear $CO_2$ molecule to form a bent $CO_2^{•-}$ anion radical (Eq. 1)[1–3]. After the rate-determining step, the intermediate $CO_2^{•-}$ is subsequently reduced via the proton-assisted electron transfer approach, in which $H_2$ or $H_2O$ is utilized as the proton source[1–3]. In these $CO_2$ reduction processes, the types of the products are determined by the number of the transferred proton–electron pairs (Eqs. 2–5)[2]:

$$CO_2 + e^- = CO_2^{•-}, \tag{1}$$

$$CO_2 + 2H^+ + 2e^- = CO + H_2O, \tag{2}$$

$$CO_2 + 4H^+ + 4e^- = HCHO + H_2O, \tag{3}$$

$$CO_2 + 6H^+ + 6e^- = CH_3OH + H_2O, \tag{4}$$

$$CO_2 + 8H^+ + 8e^- = CH_4 + 2H_2O. \tag{5}$$

Although inexpensive and abundant $H_2O$ is generally believed to be the ideal proton source[4], light-driven $CO_2$ reduction with pure water is hampered by the difficulties in accumulating photogenerated charges and appropriately coupling two half-reactions on a single catalyst[5,6]. Furthermore, the present photocatalytic $CO_2$ reductions with $H_2O$ are universally carried out in a single reactor[7,8], and the obtained gases are mixtures of hydrocarbons and $O_2$. The presence of $O_2$ inevitably leads to oxygen contamination and may also improve the oxidation of the hydrocarbons that significantly reduce the efficiency. As a result, the practical separation of $O_2$ from the mixed gas becomes a serious technological and economic issue for large-scale and sustainable applications of $CO_2$ conversion[9]. To overcome the above problems, photocatalytic $CO_2$ reductions are usually carried out by using $H_2$ as a proton source[10]. Under light irradiation, the adsorbed $H_2$ molecules are first dissociated into hydrogen adatoms ($H_2^* \rightarrow 2H^*$, where * represents the active site), which immediately react with the photogenerated holes to form $H^+$ on the catalyst surface[11]. Then, $CO_2$ reduction can be facilitated by a proton-coupled electron transfer process with a much lower potential[3]. Compared with $H_2O$, the relatively high cost and explosive feature of $H_2$ may be two drawbacks for its application in $CO_2$ conversion. Furthermore, the $H_2$ dissociation process inevitably requires an energy input, whose value depends on the used catalysts. Based on the above discussion, we can conclude that both $H_2$ and $H_2O$ have their own unique sets of advantages and shortcomings in $CO_2$ reduction. A $CO_2$ conversion strategy that can simultaneously avoid the drawbacks of $H_2$ and $H_2O$ might be revolutionary in catalytic transformation. Above all, significant opportunities for redesigning the catalytic strategy to convert $CO_2$ for practical applications are believed to exist.

Unlike traditional artificial $CO_2$ reduction by a one-step reaction, natural photosynthesis in green plants can convert $CO_2$ and $H_2O$ into carbohydrates through two sequential steps, which are known as the light and dark reactions (Fig. 1a). Under sunlight irradiation, chloroplasts can enable the synthesis of reducing equivalents (i.e., [H]) and $O_2$ from water splitting[12,13]. Then, the reducing equivalents are used to produce reduced nicotinamide adenine dinucleotide phosphate (NADPH) and adenosine triphosphate (ATP) by sequential $e^-$ and $H^+$ transfer steps, respectively. In the dark reaction, with the help of NADPH, $H^+$, and ATP, $CO_2$ reduction can be carried out stepwise to generate carbohydrates[12]. As a result, the water-splitting process is temporally and spatially separated from $CO_2$ reduction reactions by forming a reductive intermediate, which is helpful in promoting

$CO_2$ conversion by lowering the reaction barrier and avoiding charge accumulation and $O_2$ separation[14,15]. Therefore, natural photosynthesis provides a two-step reaction model for innovative catalyst design in $CO_2$ conversion with $H_2O$.

Similar to the natural photosynthesis, the decoupled approach for water splitting has been explored recently, in which water oxidation and proton reduction reactions were spatially separated[16,17]. Compared with water splitting, $CO_2$ reduction is much more difficult because $CO_2$ is a thermodynamically stable molecule with a linear structure[1]. Furthermore, finding a single material to mimic natural photosynthesis for $CO_2$ conversion is more challenging because it should work as a redox shuttle to bridge the separated water splitting and $CO_2$ reduction reactions[12,18], which is more complicated than the only water splitting process. Here, we present a design strategy to realize these sequential reactions using a single Bi catalyst (Fig. 1b). First, splitting of $H_2O$ into H atoms and $O_2$ and storage of the H atoms in Bi nanoparticles (denoted as Bi-$H_x$) were simultaneously achieved by an electrochemical approach. Then, the obtained Bi-$H_x$ was used as a reducing equivalent to reduce $CO_2$ to CO by in situ generating $H^+/e^-$ pairs under light irradiation. Meanwhile, Bi-$H_x$ recovered to Bi for reversible storage and release of hydrogen. In this process, Bi can function as an electron–proton-transfer mediator to bridge water splitting and $CO_2$ reduction reactions. A detailed mechanism for CO production with the benefits of this strategy is shown in Fig. 1b. This reaction model provides a concept for developing efficient and multifunctional catalysts for $CO_2$ conversion by mimicking photosynthesis.

## Results

Bi was loaded on nickel foam (NF) by an electrodeposition method. The NF was chosen as a substrate for depositing Bi because of its good mechanical performance and high porosity with an interconnected framework structure that is favorable for easy contact between electrolyte and electrode, and fast ion transport. As shown in Fig. 2a (red line), the diffraction peaks for Bi/NF can be assigned to the rhombohedral Bi structure (JCPDS#44-1246) and NF. The quality of the loaded Bi was determined to be 1.253 mg by inductively coupled plasma (ICP) atomic emission spectroscopy. Cyclic voltammetry (CV) curves of the porous nickel and Bi/NF electrodes in 1 M KOH electrolyte are shown in Fig. 2b. In this work, all the potentials were given by the reversible hydrogen electrode (RHE). For the NF electrode (red line in Fig. 2b), the peaks at ca. −0.03 and 0.38 V vs. RHE (vs. RHE) can be attributed to adsorption and desorption of hydrogen, respectively[19]. When Bi/NF was used as the working electrode (blue line in Fig. 2b), two cathodic adsorption/reduction and $H_2$ evolution peaks of hydrogen are obviously observed at ca. 0.17 (indicated by $H_{ads}$) and −0.04 V, respectively. During the following anodic polarization of Bi/NF, two current peaks can be observed at ca. 0.45 and 0.62 V (blue line in Fig. 2b), which are ascribed to hydrogen desorption ($H_{des}$) and hydrogen oxidation ($H_{oxi}$) on the Bi electrode, respectively[20]. Hydrogen desorption occurs before hydrogen oxidation, suggesting strong adsorption of hydrogen on the surface of Bi/NF[19]. Such behavior is not observed in the CV curve of NF in Fig. 2b, indicating that the presence of Bi is responsible for the electrochemical hydrogen storage behavior of Bi/NF. Similar hydrogen storage properties of Bi in acid solution have also been studied previously[21].

Electrochemical hydrogen storage in Bi was performed at a reduction potential of −0.18 V for 10 min, and the obtained sample is denoted Bi-$H_x$. In this process, the $H_2$ and $O_2$ evolutions by water splitting are shown in Fig. 2c. The obtained products exhibit a nearly linear rise in time from 0 to 10 min. It is worth noting that the molar ratio of $H_2/O_2$ is ~0.52, which is

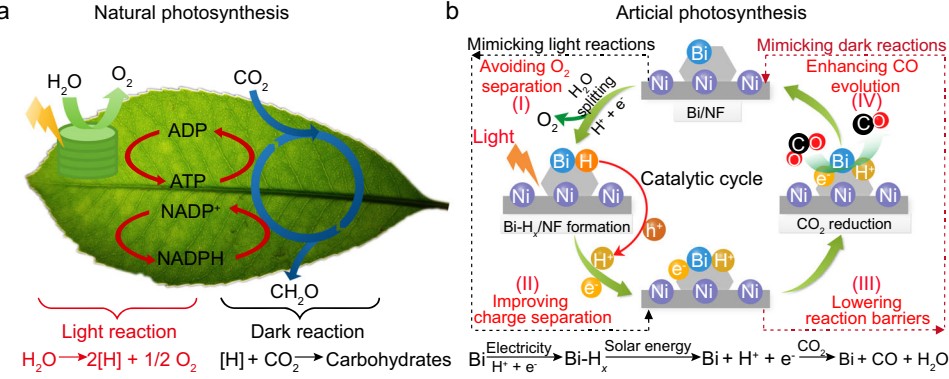

**Fig. 1 Photosynthesis in nature and the artificial analogy in this study. a** Schematic depiction of the light and dark reactions in natural photosynthesis. **b** Graphical representation of the reaction pathway of the artificial $CO_2$ reduction with the benefits of the reaction model.

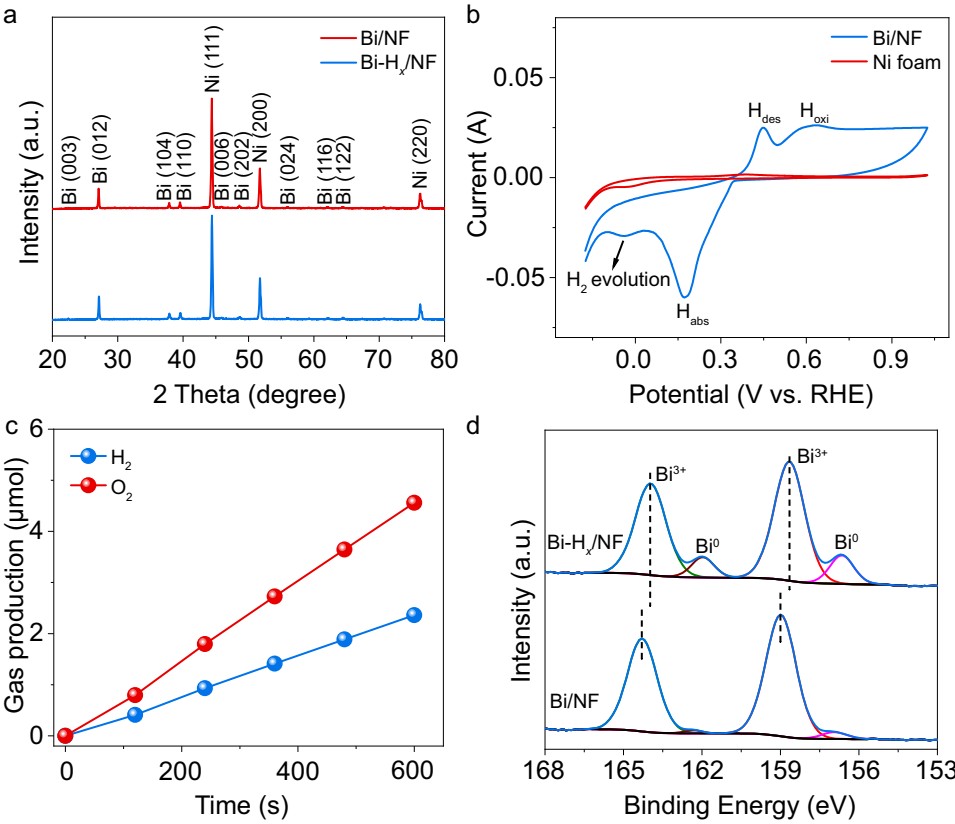

**Fig. 2 Preparation and characterization of Bi with and without stored hydrogen. a, d** XRD patterns, and high-resolution Bi $4f$ XPS spectra of Bi/NF and Bi-$H_x$/NF. **b** Cyclic voltammograms of the as-prepared Bi/NF and pristine NF in 1 M KOH solution. Scan rate: 50 mV s$^{-1}$. **c** $H_2$ and $O_2$ evolution curves on the Bi/NF electrode by water splitting at −0.18 V (vs. RHE).

much lower than the stoichiometric ratio of 2. This result suggests that a part of the reduced H species was stored in Bi electrode without desorption, which is consistent with the CV curve of Bi/NF in Fig. 2b. The number of hydrogen atoms ($M_H$) stored in Bi-$H_x$/NF was estimated to be 13.52 μmol based on the amounts of the produced $H_2$ ($M_{H_2}$) and $O_2$ ($M_{O_2}$) in Fig. 2c ($M_H = 4M_{O_2} - 2M_{H_2}$). Moreover, Faradaic efficiency (FE) of the Bi electrocatalyst for $H_2O$ splitting was calculated to be 20.9% based on the $O_2$ evolution according to Eq. (7) in the "Methods" section.

The X-ray diffraction (XRD) pattern of the Bi-$H_x$ product is shown in Fig. 2a (blue line), which is similar to that of pure Bi. This result indicates that the crystal structure of Bi was not affected by hydrogen storage. However, hydrogen storage in a metal may affect the electronic structure of the metal via electron

donation. Therefore, X-ray photoelectron spectroscopy (XPS) studies were performed to study the electronic structures of Bi/NF and Bi-$H_x$/NF. The high-resolution spectra of Bi $4f$ are presented in Fig. 2d. As shown in Fig. 2d, the XPS spectra of the two samples can be deconvoluted into four peaks related to metallic Bi and the oxidation state of Bi. The oxidation state of Bi was formed by surface oxidation during the exposure of Bi to air[22,23]. For Bi/NF, the binding energies at 159.0 and 164.2 eV are assigned to the $Bi^{3+}$ of pure $Bi_2O_3$[23]. However, for Bi-$H_x$/NF, the peaks belonging to the oxidation states of Bi are obviously red-shifted relative to those for Bi/NF, which is related to incomplete oxidation of Bi due to the electron donation of hydrogen atoms. High-resolution XPS spectra of O $1s$ for the two samples were examined to further confirm the surface chemistry of Bi

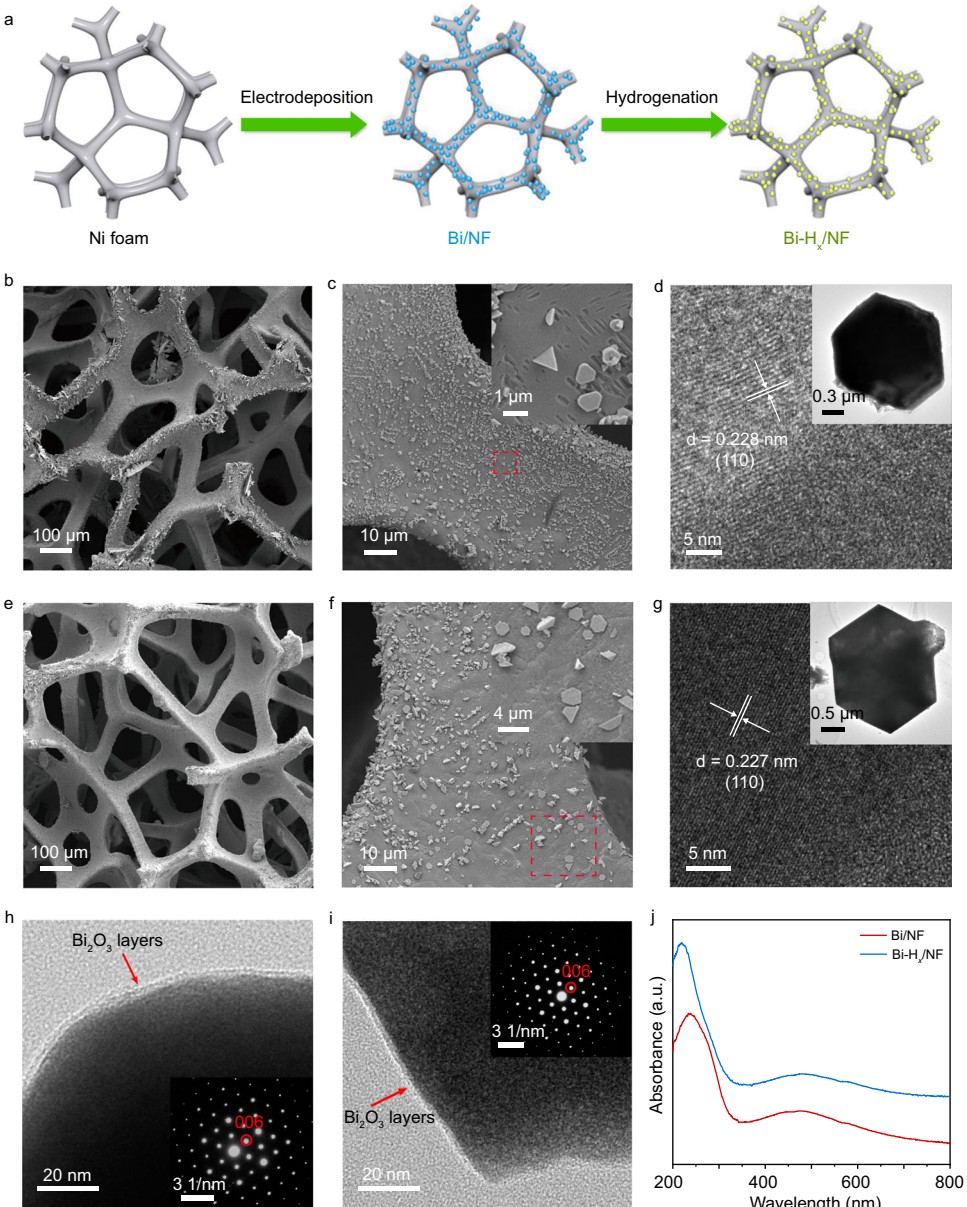

**Fig. 3 Morphologies, microstructures, and light absorption properties of the samples. a** Schematic illustration of the synthesis process of Bi-H$_x$/NF. SEM images of (**b**, **c**) Bi/NF and (**e**, **f**) Bi-H$_x$/NF. The insets in (**c**) and (**f**) are enlarged views of the corresponding SEM images marked in red squares. **d**, **g** HRTEM images of Bi/NF and Bi-H$_x$/NF. The insets in (**d**) and (**g**) are the TEM images of Bi/NF and Bi-H$_x$/NF sheets, respectively. **h**, **i** TEM images of the edges of Bi/NF and Bi-H$_x$/NF sheets for showing the formation of amorphous layers. The insets in (**h**) and (**i**) are the SEAD patterns of Bi/NF and Bi-H$_x$/NF. **j** UV–visible absorption spectra of Bi/NF and Bi-H$_x$/NF.

(Supplementary Fig. 1). The two samples exhibit similar O 1$s$ XPS spectra that can be deconvoluted into three peaks corresponding to, Bi-O bands (529.3 eV), surface hydroxyl oxygen (530.8 eV), and adsorbed O$_2$ (532.7 eV), further confirming the generation of Bi-O in Bi$_2$O$_3$ on the surface of the two samples[24]. The results of the XPS study are consistent with the fact that hydrogen storage in Bi occurs during the electrochemical treatment process. Notably, the main body of Bi/NF or Bi-H$_x$/NF is still composed of metallic Bi based on the XRD patterns in Fig. 2a.

Based on the results in Fig. 2, the loading process of Bi-H$_x$ on NF is schematically shown in Fig. 3a. The morphology of the as-prepared samples was visualized by scanning electron microscopy (SEM). The SEM images of Bi/NF and Bi-H$_x$/NF in Fig. 3b, e show that the NF has a macroporous structure with pore sizes between 100 and 400 μm. The magnified SEM images in Fig. 3c, f

indicate that both NF skeletons are covered with hexagonal sheets with a size of 0.5–2 μm, proving that Bi was successfully deposited on the surface of NF and that the hydrogen storage process did not change the morphology of Bi. The microstructures of the nanosheets were further characterized by transmission electron microscopy (TEM) and selected area electron diffraction (h). As shown in Fig. 3d, g, high-resolution TEM (HRTEM) images of the hexagonal nanosheets display crystal lattices corresponding to the (110) plane of primary Bi. As shown by the XPS spectra in Fig. 2d, the surface of Bi can be easily oxidized into Bi$_2$O$_3$ in the air. To investigate the formed Bi$_2$O$_3$ layer, the TEM analysis was performed on the edge of the Bi-H$_x$ and Bi sheets. As shown in Fig. 3h, i, amorphous Bi$_2$O$_3$ layers with a thickness of ~3 nm were clearly formed on both surfaces of Bi-H$_x$ and Bi. The SAED patterns of Bi (inset of Fig. 3h) and Bi-H$_x$ (inset of Fig. 3i) show

that only the spots ascribing to the rhombohedral Bi are observed, suggesting the single-crystalline nature of the samples. No distinct change in the microstructure is found for the sample after hydrogen storage.

Then, ultraviolet (UV)–visible absorption spectroscopy was used to study the optical properties of the samples (Fig. 3j), with the results showing that hydrogen storage does not obviously change the light absorption of Bi. As shown in Fig. 3j, the light absorption spectra of the two samples reach ~600 nm, proving that Bi/NF is a good candidate for absorbing the visible light of solar radiation. In addition, both samples display an absorption peak in the range between 400 and 600 nm. To prove that this peak is caused by the localized surface plasmon resonance (LSPR) absorptions, the Bi nanostructures with sizes of ~100 and ~400 nm were further synthesized at the deposition time of 1 and 4 min, respectively (Supplementary Fig. 2). XRD measurements show that elemental Bi was formed in these situations (Supplementary Fig. 3a). Then, the corresponding absorption spectra of the obtained Bi/NF at different deposition times were tested (Supplementary Fig. 3b). The Bi sample with a diameter of ~100 nm shows a bandgap absorption onset of ~600 nm, whereas the absorption peak at ~480 nm in Fig. 3h disappears. This optical phenomenon implies that the absorption peak between 400 and 600 nm in Fig. 3h was induced by LSPR, rather than from the coupling effect of Bi and $Bi_2O_3$ layer. Moreover, by comparing the spectra of Bi samples prepared at 4 and 10 min, we can find that the position of the absorption peak shows a red-shift with the size increasing (Supplementary Fig. 3b), which is consistent with the characteristic of LSPR absorption that is strongly correlated to the shape and size of the materials[25]. The above optical properties of the samples confirm that the absorption peak at 480 nm in Fig. 3h can be attributed to the LSPR of Bi, which is consistent with the previous report[26].

Coupling of plasmonic Bi nanoparticles can extend the light absorption of semiconductor photocatalysts and achieve enhanced activity[27]. However, direct photocatalysis on plasmonic Bi nanostructures, which is termed as plasmonic photocatalysis, has rarely been reported. Compared with the semiconductor-based photocatalysis that has received the most attention for several decades, plasmonic photocatalysis just started from the year 2011[28]. Therefore, plasmonic metallic nanostructures represent a new family of photocatalysts, and even a few reports focus on $CO_2$ reduction[29]. In plasmonic photocatalysis, a relatively high reaction temperature (>150 °C) is generally required for activating the adsorbed molecules[29–31]. Therefore, it is reasonable that the photocatalytic $CO_2$ reduction on the plasmonic Bi should be triggered by heat input. Although most plasmonic metals used as a light absorber in metal-semiconductor systems can induce photocatalytic reactions at room temperature, the low charge transfer (CT) efficiency between metals and semiconductors is the key to limit their applications[32]. Furthermore, most of the plasmonic photocatalysts for $CO_2$ reductions are based on noble metal nanoparticles, such as Ag, Au, and Rh[30,31].

After the electrochemical hydrogen storage process of Bi/NF in 1 M KOH electrolyte, the formed Bi-$H_x$/NF was used for $CO_2$ reduction in another gas–solid reactor with a quartz window on top, in which both the temperature and light illumination could be controlled. Initially, the reduction of $CO_2$ over Bi-$H_x$/NF treated in KOH for 2–12 min was investigated at 180 °C by monitoring the production of CO (Fig. 4a). Other possible products (such as $CH_4$ and $CH_3OH$) were not detected, confirming that high selectivity for $CO_2$ reduction was achieved on Bi-$H_x$/NF. As shown in Fig. 4a, the catalyst treated for 10 min shows the highest CO formation rate, and similar activity for producing CO is exhibited when the reaction time is further increased to 12 min. Therefore, we focused our study on the sample treated for 10 min

to investigate the roles of the stored hydrogen in the $CO_2$ reduction process.

Figure 4b shows the CO and $H_2$ evolution activity of Bi-$H_x$/NF at different temperatures under 420 nm light-emitting diode (LED) light irradiation with low intensity (108.6 mW cm$^{-2}$). As shown in Fig. 4b, 100% selectivity towards CO production (99.0 μmol g$^{-1}$ h$^{-1}$) is achieved on Bi-$H_x$/NF at 150 °C. When the reaction temperature reaches 160 °C, $H_2$ evolution is observed, which increases with increasing temperature. Enhancing the reaction temperature can induce an increase in the relative population of adsorbed $CO_2$ molecules in excited vibrational states based on the Bose–Einstein distribution[33]. Due to this effect, a higher temperature is required to facilitate activation and subsequent reduction of $CO_2$ molecules, as shown in Fig. 4b. Because no proton sources were added into the reactor, the $H_2$ evolution behavior should be caused by the release of $H_2$ from Bi-$H_x$/NF upon heating, which is easy to understand according to the simple Le Chatelier's principle. The desorption of $H_2$ molecules from metals with stored hydrogen induced by the heating effect has been extensively studied experimentally[34]. As shown in Fig. 4b, the CO production rate increases with temperature and reaches a maximum of 283.8 μmol g$^{-1}$ h$^{-1}$ at 180 °C. Significantly, this high CO production rate on the simple Bi catalyst is achieved without the use of any cocatalysts. Therefore, we can conclude that the two-step strategy holds great potential in $CO_2$ reduction with $H_2O$. As shown in Fig. 4b, a further increase in temperature reduces the CO evolution rate because many more reactive H atoms are converted to $H_2$ molecules. The $H_2$ release event further demonstrates that H atoms were indeed stored in the Bi nanocrystals, resulting in the formation of a reductive surface on Bi that can cause $CO_2$ reduction.

As a control experiment, pristine NF was treated at a reduction potential of −0.18 V for 10 min in 1 M KOH electrolyte and then placed in the gas–solid reactor for testing the thermo-photocatalytic $CO_2$ reduction activity. Under identical reaction conditions to Bi-$H_x$/NF, no CO production is detected after 4 h of reaction, indicating a critical role of Bi-$H_x$ in driving photocatalytic $CO_2$ reduction. Control experiments in the absence of light, heat, $CO_2$, or catalyst were further performed, and the CO product is not detected, demonstrating that CO production on Bi-$H_x$/NF is driven by $CO_2$ reduction under the synergetic effect of photon flux and thermal energy.

Considering that the heat input can be provided by the infrared light of sunlight (accounting for ~50% of the solar energy), thus the plasmonic Bi photocatalysts can potentially use the entire solar spectrum for driving catalytic reactions by employing a high-temperature solar reactor driven by concentrated solar radiation. Although conventional semiconductor photocatalysts can work at room temperature, their activities always decreased with increased temperatures due to the relatively low Debye temperatures of the semiconductors[30,35]. Therefore, in practical applications, the heat effect from concentrated sunlight might be one of the hindrances for efficient solar energy conversion in gas–solid reaction system by using semiconductor photocatalysts[30]. In this respect, the ability of the plasmonic Bi photocatalyst that can work at 180 °C might have its own advantage for $CO_2$ reduction.

Moreover, an isotopic $^{13}CO_2$ labeling experiment was also carried out on Bi-$H_x$/NF under identical reaction conditions (Fig. 4c) on the basis of gas chromatography (GC)-mass spectrometry. The peak attributed to the Ar gas was not provided in Fig. 4c because the high intensity of it disturbed the distinguishing of other low-intensity peaks. A peak is observed at $m/z$ = 29 ($^{13}CO$), further confirming that the generated CO originates from the light-induced reduction of $CO_2$ rather than from contaminants[7]. For comparison, the isotopic $^{12}CO_2$ labeling

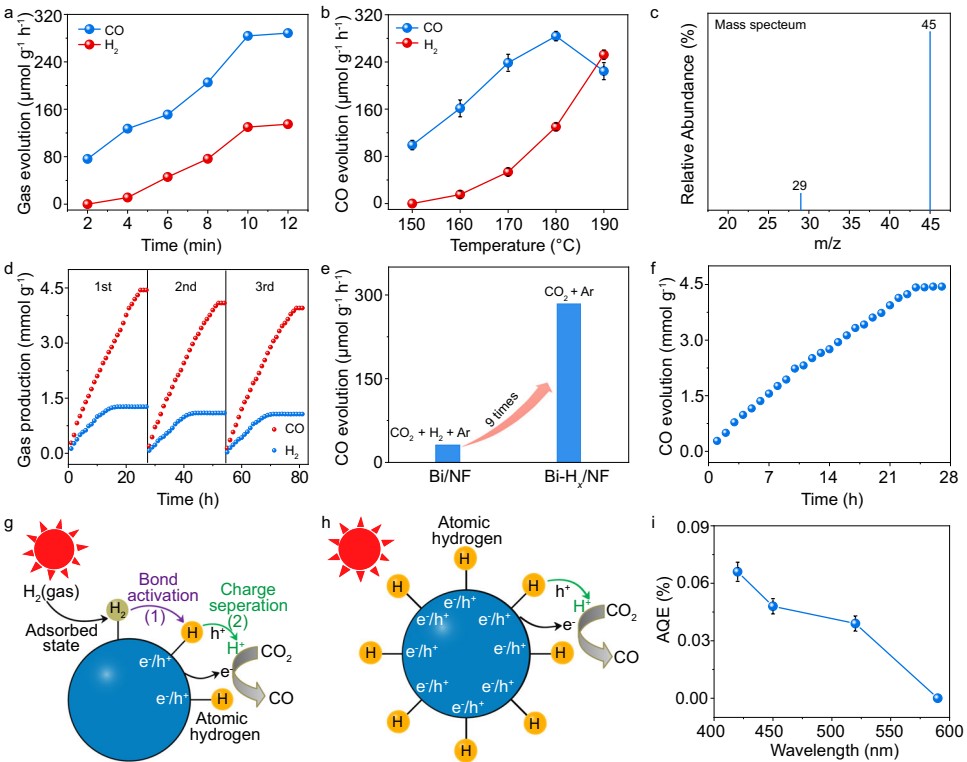

**Fig. 4 Thermo-photocatalytic CO evolution tests. a** Reduction of $CO_2$ over Bi-$H_x$/NF catalysts treated in KOH for different times. **b** Effect of temperature on CO and $H_2$ evolution rates over Bi-$H_x$/NF under a 420-nm LED light irradiation ($n = 3$, error bars: standard deviation). **c** Mass spectrum from GC-MS analysis of the CO generated in the catalytic $CO_2$ reduction reaction using $^{13}CO_2$. **d** Stability test of Bi-$H_x$/NF using three repeats of the $CO_2$ reduction reaction under 420-nm LED light irradiation at 180 °C. **e** Comparison on the CO production activities between Bi/NF using $H_2$ as proton source and Bi-$H_x$/NF using the stored H as a proton source. **f** CO evolution curve on the Bi-$H_x$/NF catalyst treated in the air at 80 °C for 0.5 h. **g, h** Schematic diagram of $CO_2$ reduction processes using $H_2$ as proton source on Bi/NF and preformed H species as proton source on Bi-$H_x$/NF. **i** AQEs of CO evolution over the Bi-$H_x$/NF catalyst under different wavelength irradiation ($n = 3$, error bars: standard deviation).

experiment was also performed on Bi-$H_x$/NF (Supplementary Fig. 4), and $^{12}$CO with $m/z = 28$ is found to be the main product for $^{12}CO_2$ reduction, which further demonstrates that CO was produced from the thermo-photocatalytic reduction of $CO_2$ on Bi-$H_x$/NF. As shown in Fig. 4c, except for the peaks belonging to $CO_2$ and CO, no other peaks were observed in Fig. 4c, further confirming the high selectivity for $CO_2$ reduction on Bi-$H_x$/NF.

To prove the recycling capacity of Bi-$H_x$/NF, repeated $CO_2$ reduction experiments under 420-nm LED light irradiation were carried out (Fig. 4d). After each run, the Bi-$H_x$/NF was rehydrogenated by the same electrochemical method (reacting in 1 M KOH solution at a voltage of −0.18 V for 10 min) and added into the reactor with $CO_2$ gas. As shown in Fig. 4d, the CO evolution reaction stops after nearly 27 h of reaction in each run because the surface of Bi-$H_x$/NF loses the ability to reduce $CO_2$ with the consumption of the stored hydrogen atoms. In this process, Bi-$H_x$ gradually recovers to Bi, which can be further used for H storage and CO production, as shown in the second and third runs in Fig. 4d. This observation indicates that the stored hydrogen indeed plays an important role in $CO_2$ reduction. In the third run, the CO evolution on Bi-$H_x$/NF can still maintain ~90% of the initial activity after 81 h of reaction, suggesting that Bi-$H_x$/NF is stable and that $CO_2$ reduction on Bi-$H_x$ can be achieved by a catalytic cycle. The XRD pattern and an SEM image of Bi-$H_x$/NF after 81 h of photocatalytic reaction were obtained (Supplementary Fig. 5), and no apparent changes in the crystal structure or morphology are observed, indicating that Bi-$H_x$/NF has favorable chemical stability. These results confirm that Bi/NF can function as a reversible hydrogen storage material to complete the catalytic cycle.

It is well known that $H_2$ is traditionally used as the proton source for photochemical $CO_2$ conversion via the proton-assisted electron transfer approach[3]. In order to show that the current approach using surface-bound H atoms as a proton source has advantages over the traditional $H_2$, the controlled experiment for $CO_2$ reduction was carried out on Bi/NF material in dissociative $H_2$ gas (0.5 mL). As shown in Fig. 4e, although the number of hydrogen atoms stored in the Bi-$H_x$/NF systems (13.52 µmol) is only ~25% of that from $H_2$ in Bi/NF reaction system, the $CO_2$ reduction activity of Bi-$H_x$/NF reaches nine times that of Bi/NF. This result indicates that Bi-$H_x$/NF has great utility for achieving high activity in catalytic $CO_2$ reduction by concentrating reactive hydrogen adatoms on the catalyst.

As shown in the XPS spectra in Fig. 2d, the amount of Bi[0] species on the surface of Bi-$H_x$/NF is larger than that of Bi/NF. In order to exclude that the enhanced activity of Bi-$H_x$/NF in Fig. 4e is not caused by the increased content of Bi[0], the Bi[0] on the surface of Bi-$H_x$/NF was oxidized into Bi[3+] by treating the sample in the air at 80 °C for 0.5 h. The absence of Bi[0] species on the surface of Bi-$H_x$/NF after the treatment can be proved by XPS measurement (Supplementary Fig. 6). In the following, the controlled experiment for $CO_2$ reduction was carried out on the sample after heat treatment. Compared with the activity of Bi-$H_x$/NF in Fig. 4d, no performance degradation on CO production was found on Bi-$H_x$/NF when the surface Bi[0] species was replaced by Bi[3+] (Fig. 4f), indicating that the improved activity of Bi-$H_x$/NF in Fig. 4e cannot be attributed to the increased amount of surface Bi[0].

Based on the above results, it can also speculate that the automatically formed $Bi_2O_3$ layer on Bi-$H_x$/NF in the air plays

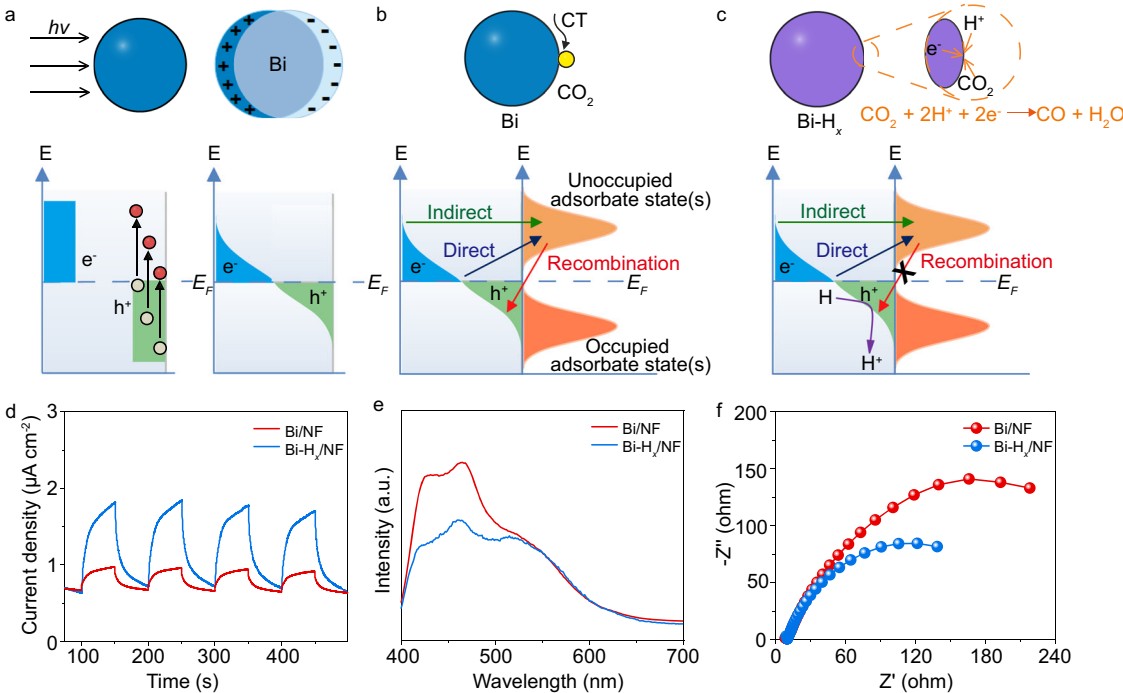

**Fig. 5 Effects of hydrogen storage on the photoinduced CT property of Bi. a** Generation and relaxation of LSPR-induced carriers under light irradiation. The blue areas above the Fermi energy ($E_F$) and the green areas below $E_F$ represent the distributions of excited electrons and holes, respectively. **b** Schematic illustration of the transfer pathways of the LSPR-induced carriers in Bi through direct and indirect CT processes. **c** Graphical representation of the effect of the H stored in Bi-$H_x$ on improving the carrier separation in the $CO_2$ reduction reaction. **d** Photocurrent transient responses of Bi/NF and Bi-$H_x$/NF catalysts under a 420-nm LED light irradiation at a bias of 1.07 V (vs. RHE). **e** The steady-state PL spectra of Bi/NF and Bi-$H_x$/NF. **f** Nyquist plots of the Bi/NF and Bi-$H_x$/NF catalysts at a bias of 1.07 V (vs. RHE).

negligible effect on the $CO_2$ reduction activity. For the Bi-$H_x$ covered by ~3 nm amorphous $Bi_2O_3$ layer, tunneling of the hot carrier to the out surface is still feasible due to the high energy of the hot charge carriers generated by LSPR and a high-density of defect states in the amorphous layer. A similar result has been observed on Bi catalysts for electrochemical $H_2$ production and $CO_2$ reduction in previous reports[22,24]. In addition, the specific surface areas of the two samples were studied by the Brunauer–Emmett–Teller (BET) method based on the nitrogen adsorption isotherm, showing that the BET surface areas of Bi-$H_x$ and Bi are 3.3356 and 2.1039 $m^2\,g^{-1}$, respectively. This result demonstrates that the surface area should not be the main factor for the enhanced $CO_2$ reduction activity of Bi-$H_x$/NF.

For the $CO_2$ reduction process on Bi/NF with $H_2$ gas, the photocatalytic interface must facilitate two sequence steps: (1) the adsorption and decomposition of the $H_2$ to generate atomic H bound on the surface, and (2) the transfer of these adsorbed H atoms into electron–proton pairs to reduce $CO_2$ (Fig. 4g). However, the two coupled steps (1 and 2) in both time and space preventing independent optimization of each, and the competitive adsorption of $H_2$, may impede $CO_2$ activation at the active sites[36]. Moreover, the decomposition of the $H_2$ in step (1) will inevitably require certain energy input. By separating the above two steps (1 and 2), the present approach for $CO_2$ conversion addresses the incompatibilities of functions (1) and (2) and bypasses the $H_2$ decomposition process in the traditional $CO_2$ reduction process with $H_2$ (Fig. 4h). The improved $CO_2$ reduction activity on Bi-$H_x$/NF can be attributed to the atomic H atoms preformed on the surface, which not only improve photoinduced charge separation, but also lower the $CO_2$ reduction barriers. These two positive effects will be closely discussed in the following Figs. 5 and 6, respectively.

To evaluate the effect of light on the catalytic CO evolution, the apparent quantum efficiency (AQE) as a function of the irradiation wavelength on Bi-$H_x$/NF was examined, in which thermal contribution was considered in the calculation of AQEs (see details in "Methods" section). As shown in Fig. 4i, the AQEs at 420, 450, and 520 nm reach 0.066%, 0.048%, and 0.034%, respectively, confirming that the plasmonic excitation of Bi-$H_x$/NF plays an important role in $CO_2$ reduction. Obviously, the plasmonic photocatalyst of Bi-$H_x$/NF is a type of catalyst that can combine light and thermal energy together for exciting $CO_2$ reductions. To objectively understand the quantum efficiency of the plasmonic Bi catalysts, examples of thermal $CO_2$ hydrogenation on metals, photocatalytic $CO_2$ reduction based on plasmonic metals, and plasmonic metal coupled with semiconductor systems are compared. Although a temperature of 180 °C is needed in the present Bi-$H_x$/NF system, this temperature is much lower than the thermochemical $CO_2$ conversion process on metal catalysts (550–750 °C)[37]. Recently, plasmonic photocatalysis based on noble metal nanostructures was reported for $CO_2$ reduction with a higher activity than the present Bi catalyst[30,31]. It should be indicated that the high activities on the noble metals were obtained under intense light illuminations (above 1 $W\,cm^{-2}$). However, the present Bi catalyst can induce $CO_2$-to-CO conversion with a relatively high CO evolution rate of 283.8 $\mu mol\,g^{-1}\,h^{-1}$ under low-intensity LED light illumination (108.6 $mW\,cm^{-2}$, around solar intensity). Moreover, in comparison to most semiconductor systems coupled with plasmonic metals, the present catalytic circle for $CO_2$ reduction shows a higher photocatalytic activity, although some semiconductors with significantly high efficiencies were reported[1,4,29,38,39]. More details on the photocatalytic $CO_2$ reduction efficiencies on semiconductor systems can be found in the recent review articles[1,4]. Rather than the catalytic efficiency, the main point of

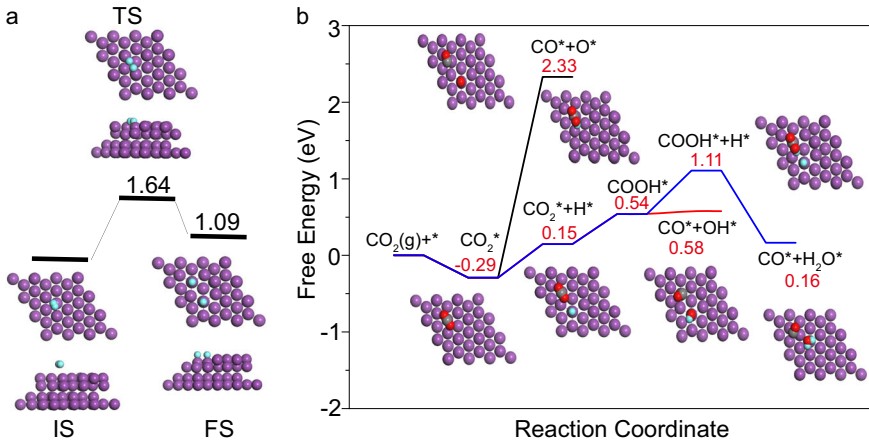

**Fig. 6 DFT calculations. a** Reactant, transition state, and product of $H_2 \rightarrow 2H^*$ on Bi. **b** Free energy diagrams of $CO_2 \rightarrow CO$ on Bi with (blue and red lines) and without (black line) hydrogen storage. The purple, gray, red, and blue balls represent Bi, C, O, and H atoms, respectively.

the present manuscript is to demonstrate a two-step reaction model for $CO_2$ conversion by using a plasmonic Bi metal that entirely relies on nonprecious materials.

Recently, a growing number of plasmonic photocatalysts for driving chemical reactions by producing hot carriers under visible light illumination have been reported[40]. According to previous reports[41,42], two possible mechanisms (indirect and direct) in chemical reactions driven by plasmonic metals are schematically shown in Fig. 5a, b. In the indirect mechanism, hot electron–hole pairs are generated in metal nanoparticles under light irradiation (Fig. 5b), forming a Fermi–Dirac-type distribution of hot electrons. In this distribution, hot electrons with sufficient energy can transfer to the unoccupied states of the adsorbed molecules, forming transient negative ions that then decay back to the metal. In the direct mechanism, electrons are assumed to be directly injected from the metal into the unoccupied state of the adsorbed reactant, leaving holes in the plasmonic metal (Fig. 5b). Compared with the indirect pathway, the formation of occupied states within the metal is avoided in the direct mechanism. Although some ramifications exist regarding the underlying mechanism in plasmonic photocatalysis, approaches that can improve the separation rates of hot electron–hole pairs should be beneficial for the charge injection process in both the indirect and direct mechanisms[41,42].

Compared with the photogenerated charge in semiconductors, the lifetime of hot electrons produced by LSPR is much shorter[40]. Therefore, improving the separation efficiency of hot charges in plasma metals is particularly important. Similar to semiconductor photocatalysts, the electron transfer reactions on plasmonic Ag nanoparticles can be improved by using hole scavengers[41,42]. This phenomenon demonstrates that charge scavengers can also play an important role in driving photocatalytic reactions on plasmonic metals. Consequently, the role of the stored H atoms in the $CO_2$ reduction of Bi-$H_x$/NF is discussed based on the above consideration (Fig. 5c). Under light irradiation, hot electron–hole pairs are formed on irradiated Bi-$H_x$ nanosheets through plasmonic excitation, as shown in Fig. 5a. The plasmonic hot holes are consumed by the H atoms stored in Bi-$H_x$/NF ($H + h^+ \rightarrow H^+$). In one operation, reactive $H^+$ and $e^-$ are synchronously generated on Bi-$H_x$/NF and react with the activated $CO_2$ molecules on the active sites (Fig. 5c). Based on this mechanism, suppression of the recombination of photoinduced carriers and enhancement of electron and proton donation are simultaneously realized by prepositioning H in the Bi catalyst, which accounts for the superior $CO_2$ reduction performance on Bi-$H_x$/NF through the proton-assisted electron transfer approach ($CO_2 + 2H^+ + 2e^- \rightarrow CO + H_2O$). Furthermore,

the interdependent nature of $H^+$ and $e^-$ avoids the accumulation of $H^+$ or $e^-$ in the reaction process[15], which is also beneficial for improving the performance of Bi-$H_x$/NF. The fascinating point of Bi-$H_x$/NF lies in the bifunctional H atoms. (1) They can act as hole scavengers to promote charge separation. (2) The produced protons are subsequently used as reactants that participate in the $CO_2$ reduction reaction.

To provide evidence for the enhanced separation of photo-generated charge carriers on Bi-$H_x$/NF, photoelectrochemical measurements were carried out. Figure 5d shows the transient photocurrent curves of the samples before and after hydrogen storage during four on-off cycles. As shown in Fig. 5d, the photocurrent intensity of Bi-$H_x$/NF (0.95 $\mu A\ cm^{-2}$) is approximately four times that of pristine Bi/NF (0.23 $\mu A\ cm^{-2}$), indicating that a much lower recombination rate of photoinduced electrons and holes is indeed achieved by introducing hydrogen into Bi. To further study the separation of photoinduced electron–hole pairs under light illumination, steady-state photoluminescence (PL) analysis was performed on the two samples. As shown in Fig. 5e, the PL intensity of Bi-$H_x$/NF is significantly decreased compared with that of Bi/NF, indicating that the incorporation of H atoms on Bi/NF has effectively suppressed the radiation recombination of charge carriers, which is helpful for $CO_2$ reduction reaction on Bi-$H_x$/NF. In addition, electrochemical impedance spectroscopy was used to investigate the CT resistance of the samples in the dark. The semicircular Nyquist plots are shown in Fig. 5f. Compared with Bi/NF, a much smaller semicircle diameter is detected for Bi-$H_x$/NF, reflecting that the CT resistance is greatly reduced by storing hydrogen in Bi. The enhanced charge separation and transfer efficiencies play a role in facilitating photoinduced CO evolution on Bi-$H_x$/NF.

In order to experimentally prove that the thermal-photocatalytic $CO_2$ reduction on Bi-$H_x$ was performed by the proton-assisted electron transfer approach, the following experiments were designed. First, $CO_2$ reduction on Bi-$H_x$/NF was performed at 180 °C without light illumination by using $H_2O$ (200 μL) as a proton source. The negligible CO production activity (Supplementary Fig. 7) demonstrates that photo-induced charges are the primary factor for $CO_2$ reduction on Bi-$H_x$/NF. Second, under the thermal-photocatalytic conditions, the formation of negligible CO on the bare Bi/NF without using proton sources (Supplementary Fig. 8) confirms that the $CO_2$-to-CO conversion can hardly occur only with hot-electron transfer into unoccupied orbitals of adsorbed $CO_2$. Based on the above two experiments, we can conclude that the thermal-photocatalytic CO production performance on Bi-$H_x$/NF should originate from the

synergetic effect between $H^+$ and hot $e^-$, which is accordant with previous reports[1–3]. The above experiments, taken together, provide solid evidence that the proton-assisted electron transfer approach indeed took place in the $CO_2$ reduction process of Bi-$H_x$/NF as shown in Fig. 5a–c, which is consistent with the density functional theory (DFT) calculations in Fig. 6.

To further understand the reaction mechanism, the effect of irradiation intensity on the thermal-photocatalytic $CO_2$ reduction activity of Bi-$H_x$/NF was examined (Supplementary Fig. 9). The strong dependence of the evolution rate of CO on the light intensity agrees well with the proposed mechanism that the synchronous production of reactive $H^+$ and $e^-$ pairs in Bi-$H_x$ was achieved by the consumption of photogenerated holes by the stored H atoms. Moreover, the XPS curves of Bi-$H_x$/NF after different thermo-photocatalytic reaction times were measured (Supplementary Fig. 10). It can be seen that the XPS band ascribed to the oxidation state of Bi progressively blue-shifts with increasing reaction time, which is consistent with the fact that the consumption of H species proceeds during the $CO_2$ reduction reaction on Bi-$H_x$/NF. As a result of the depletion of H species after 27 h reaction, the oxidation state of Bi in Bi-$H_x$/NF returned to the state that is similar with Bi/NF without H storage (Supplementary Fig. 10). The reaction time-dependent XPS spectra also agree well with the proton-assisted electron transfer mechanism for $CO_2$ reduction on Bi-$H_x$/NF. In addition, based on the XPS analysis of Bi-$H_x$/NF, no obvious change in the ratio of $Bi^0$ and $Bi^{3+}$ is observed after 27 h reaction, indicating the good stability of the catalyst under thermo-photocatalytic reaction conditions.

Experimental studies on $CO_2$ reduction have shown that hydrogen storage in Bi plays an essential role in improving CO production compared with Bi in $H_2$ gas (Fig. 4e). To further understand the difference between the performance of Bi-$H_x$/NF and Bi/NF, DFT calculations were carried out. As discussed in the "Introduction" section, dissociation of $H_2$ into hydrogen atoms is necessary for $CO_2$ reduction. Therefore, the $H_2$ dissociation pathway on Bi(001) was first calculated, and the potential energy curve is shown in Fig. 6a. The dissociation kinetic barrier (1.64 eV) for $H_2$ on the pristine Bi(001) surface is high, which suggests that $H_2$ dissociation is difficult in terms of kinetics. Furthermore, according to the calculation results, $H_2$ dissociation is an endothermic process with an energy of 1.09 eV (Fig. 6a). Therefore, we can conclude that $H_2$ decomposition is both kinetically and thermodynamically unfavorable on the Bi surface. Compared with the photoreduction of $CO_2$ on Bi in $H_2$ gas, the $CO_2$ conversion on Bi-$H_x$ avoids the unfavorable $H_2$ dissociation step, and the stored H might directly react with $CO_2$.

To further understand the role of the stored hydrogen in the $CO_2$ reduction process, we calculated the free energy diagrams of $CO_2$ reduction pathways on Bi/NF and Bi-$H_x$/NF. The free energy curves of these pathways are depicted in Fig. 6b. As shown in Fig. 6b, direct decomposition of $CO_2$ ($CO_2 \rightarrow CO + O$) with a high thermodynamic barrier of 2.33 eV occurs on the Bi surface. In contrast, if hydrogen-stored Bi (Bi-$H_x$/NF) is used as the catalyst, then $CO_2$ reduction will proceed via a proton-assisted electron transfer approach. In this process, one more reaction pathway of $CO_2^* + H^* \rightarrow COOH^*$ occurs. The thermodynamic energy barrier in this process is calculated to be 0.58 eV, which is lower than that of $CO_2$ directly dissociating on the Bi surface, suggesting that atomic hydrogen plays a key role in the enhancement of the catalytic activity for $CO_2$ reduction. Therefore, in addition to avoiding the $H_2$ decomposition process, the storage of hydrogen would promote $CO_2$ reduction on Bi-$H_x$/NF by lowering the Gibbs free energy of the reaction pathway. According to the above DFT calculations, the hydrogen stored in Bi-$H_x$/NF is further demonstrated to be responsible for the improved performance in the catalytic

reduction of $CO_2$, consistent with the experimental results shown in Fig. 4e.

As shown in the XPS spectrum of Bi-$H_x$/NF in Fig. 2d, a part of Bi on the surface of Bi-$H_x$/NF can be oxidized in the air. To clarify the effect of the oxidation state of Bi on $CO_2$ reduction, DFT studies on oxidized Bi(001) were performed. As shown by the potential energy curves for $H_2$ dissociation pathways on the oxidized Bi(001) surface (Supplementary Fig. 11a), the $H_2$ molecule can be directly dissociated into atomic H* on oxidized Bi(001) with an energy gain of 1.30 eV and a large barrier of 1.86 eV, similar to that on the pristine Bi(001) surface. In addition, the free energy profiles associated with the reaction pathways of $CO_2$ with and without stored hydrogen on the oxidized Bi (001) surface were calculated (Supplementary Fig. 11b). By comparison with the results in Fig. 6b, we can conclude that the energy profiles are not dramatically affected by the presence of Bi oxide. Consequently, the effect of oxidized Bi on the $CO_2$ reduction activity of Bi-$H_x$/NF and Bi/NF can be neglected.

## Discussion

Based on the above results, the entire catalytic reaction on Bi/NF is schematically shown in Fig. 1b, emphasizing that the reduction of $CO_2$ in the present system occurs via two sequential steps: water splitting followed by $CO_2$ reduction reactions. In the first step, electrochemical water splitting on Bi/NF leads to the simultaneous formation of $O_2$ and Bi-$H_x$/NF (as the reduced form of Bi/NF), similar to the light reaction in natural photosynthesis. In the second step, the reduction of $CO_2$ to CO occurs on the reductive Bi-$H_x$/NF by producing $H^+/e^-$ pairs under light irradiation, similar to the dark reaction in photosynthesis. After this reaction, the Bi-$H_x$ recovers to Bi, which can be further used for storing and releasing H. In this strategy, the spatially separated water splitting and $CO_2$ reduction reactions are integrated by using Bi as an electron–proton-transfer mediator. As a result, catalytic overall splitting of $CO_2$ on Bi/NF is realized by the two sequential reactions, which correspond to the light and dark reactions in natural photosynthesis (Fig. 1a). Therefore, the whole $CO_2$ conversion on Bi/NF with the help of $H_2O$ can be regarded as artificial photosynthesis. In fact, the water-splitting process can be powered by renewable electricity (e.g., solar and wind). Consequently, the catalytic cycle on Bi/NF for overall $CO_2$ splitting can be driven by renewable energy.

It should be indicated that the low FE of the Bi catalyst (20.9%) in the decomposition of $H_2O$ is a drawback for it. However, the present manuscript does not aim to develop high-efficiency electrocatalysts, but rather demonstrates a two-step reaction model mediated by plasmonic Bi-$H_x$ for $CO_2$ conversion by mimicking photosynthesis. Improving the electrochemical performance of Bi is important for reducing the overall energy input in the present two-step $CO_2$ reduction process, which is the subject of ongoing investigations in our laboratory.

Although electrocatalytic reduction of $CO_2$ into formate ($HCOO^-$) on Bi electrode can reach a much higher FE efficiency than the electrochemical $H_2O$ splitting on the present Bi/NF, higher overpotentials between $-0.67$ and $-0.87$ V (vs. RHE) were used in previous reports[43–45]. Furthermore, the direct reduction of $CO_2$ by the electrochemical approach is not a perfect technology because it still suffers from low product concentrations in the mixture of traditional liquid electrolytes[46]. As a result, the product cannot be directly used without further purification. However, the purification of the low concentration liquid fuel in electrolytes not only compromises energy efficiency, but also is a technical challenge[46].

It is well known that different from acidic form, formate has no obvious usage[47]. In contrast, CO gas is a critical feedstock for

directly synthesizing a variety of chemicals in industry[47]. Therefore, compared with electrochemical $CO_2$ reduction on Bi, the $CO_2$-to-CO conversion achieved on the present Bi catalyst has great potential in practical applications. Furthermore, the electrocatalytic approach displays poor solubility for gaseous substrates and limits the rate of the substrate bond activation. Therefore, it is difficult to perform direct $CO_2$ reduction in the electrocatalytic approach[36]. In addition, although the gas product and $O_2$ generation in electrocatalytic $CO_2$ conversion could also be spatially separated by a membrane, the used membrane is expensive and tends to be degraded at low current densities[17]. However, the above three drawbacks do not exist in the present catalytic system for $CO_2$ reduction. Based on our understanding, there is a noticeable absence of a consummate material that can solve all of these challenges in $CO_2$ reduction together.

In addition to using $H_2$ and $H_2O$ as proton sources, developing hydrides as hydrogen storage materials are believed to be an emerging field for the reduction of $CO_2$. As attractive candidates, hydrides of silicon (Si) nanostructures have been sufficiently studied for converting $CO_2$ to useful chemicals, such as formaldehyde, CO, and methanol[48–51]. In these processes, the Si hydrides were used as reducing agents to react stoichiometrically with $CO_2$. However, achieving the catalytic conversion of $CO_2$ on Si hydrides is difficult because the formation of inactive Si–OH and Si–O–Si groups after $CO_2$ reduction prevents recovery of Si surface hydrides[52]. Although catalytic $CO_2$ conversion on silicon–hydride nanosheets was realized, the use of $H_2$ gas as a reductive agent and loading of precious palladium nanoparticles are simultaneously needed[52]. Our $CO_2$ conversion strategy is important because overall $CO_2$ splitting with water is achieved by using a low-cost, environmentally friendly Bi catalyst as a hydrogen shuttle to bridge water splitting and $CO_2$ reduction reactions, which provides an effective strategy for designing $CO_2$ reduction catalysts by separately mimicking the light and dark reactions in natural photosynthesis. Essentially, the two-step strategy successfully enables reactive hydrogen adatoms to concentrate on the active site, which provides a high driving force for $CO_2$ reduction on Bi-$H_x$/NF. More importantly, in the artificial photosynthesis, the $O_2$ separation procedures encountered in the traditional overall $CO_2$ splitting systems are successfully avoided.

## Methods

The Bi-$H_x$ sample was loaded on NF by electrochemical method. Prior to Bi-$H_x$ loading, NF (thickness: 1.0 mm, area: $1 \times 4.5$ cm$^2$) was cleaned in acetone, hydrochloric acid (0.1 mol L$^{-1}$), and deionized water under ultrasonication for 20 min each. Then, it was dried by flowing air. Electrodeposition of Bi nanosheets on the NF was carried out in a standard three-electrode system containing an aqueous solution of 0.013 M Bi(NO$_3$)$_3$ and 1 M HNO$_3$. The NF, platinum wire, and a Ag/AgCl electrode were used as the working electrode, counter electrode, and reference electrode, respectively. The synthesis of Bi/NF was performed at a reduction potential of 0.16 V (vs. RHE) for 15 min. To synthesize Bi-$H_x$/NF, Bi/NF was treated in the above system using KOH (1 M) as the electrolyte for 2–12 min at a voltage of $-0.18$ V (vs. RHE). In this process, the generated $O_2$ gas was quantified by Neofox-GT oxygen probe (Ocean optics). Finally, the obtained Bi-$H_x$/NF electrodes were immediately washed with deionized water and then dried at 80 °C for 2 h in Ar gas.

The content of Bi loaded on NF was analyzed by ICP. XRD patterns were recorded on a D8 Advance Bruker diffractometer with Cu Kα radiation. Data were collected from 20° to 80° in $2\theta$ with a scan rate of 0.02° steps s$^{-1}$. The morphology of the prepared samples was characterized by SEM (JSM-6510 microscope). TEM and HRTEM images were acquired on a TEM (JEM 2100F). XPS data were measured on a Kratos AXIS Supra spectrometer using monochromatic AlKα radiation as the excitation source. All binding energies were referenced to the C1s peak (284.8 eV) arising from adventitious carbon. UV–visible absorption spectra were recorded on a Shimadu UV-2600 spectrometer. PL spectra were obtained on a fluorescence spectrophotometer (F-7000, Hitachi, Japan) at room temperature. The BET surface area was determined by nitrogen adsorption on a Micromeritics ASAP 2460 nitrogen adsorption apparatus. An HP 5973 GC-mass spectrometer was employed to analyze the $^{13}$CO generated from the $^{13}$CO$_2$ isotopic experiment. Electrochemical measurements were performed using an electrochemical analyzer (CHI660E, Chenhua, Shanghai) with a standard three-electrode system in the normal atmosphere. The prepared sample, Pt wire, and a saturated Ag/AgCl electrode were used as the working electrode, counter electrode, and reference electrode, respectively. CV curves and Nyquist plots were obtained at a scan rate of 50 mV s$^{-1}$ in 1 M KOH aqueous solution. The observed potentials were converted to RHE based on:

$$E_{RHE} = E_{Ag/AgCl} + 0.197 + 0.059 \times pH. \quad (6)$$

FE of $O_2$ was calculated through:

$$FE = znF/Q, \quad (7)$$

where $z$ is the number of electron transfer, $n$ is the total moles of $O_2$, $F$ is the Faraday constant of 96485 C mol$^{-1}$, and $Q$ is the total amount of electricity.

$CO_2$ reduction reactions of the synthesized samples were carried out in a 300 mL closed gas system with a quartz window on top. A 420-nm LED light was employed as the light source. The temperature was measured by a thermocouple and controlled by a heating source. The Bi-$H_x$/NF sample was cut into a rectangular shape of $1 \times 1$ cm$^2$ for activity tests. The $CO_2$ reduction reactions were carried out in a gas mixture of $CO_2$ and Ar. The mole ratio of $CO_2$:Ar was 1:5. For detecting CO and $CH_4$, the gas products on Bi-$H_x$/NF were analyzed by a GC (Agilent 7890B) with a thermal conductivity detector. The GC was equipped with a 5 Å molecular sieve column using He as the carrier gas. For detecting $CH_3OH$, the gaseous products from the reactor were analyzed by an offline GC (Fuli Corp., China) equipped with a flame-ionized detector. The equipped column was TDX-01. For comparison, $CO_2$ reduction reactions on Bi/NF were also performed in a gas mixture of $CO_2$ (50 mL), $H_2$ (0.5 mL), and Ar (249.5 mL), with the other conditions unchanged.

The AQEs for CO evolution on Bi-$H_x$/NF were measured under the same catalytic reaction conditions with irradiation of different wavelengths (420, 450, 520, and 590 nm). The light intensity ($W_l$) was monitored by a PL-MW200 photoradiometer (Perfect Light, Beijing, China). Thermal contribution at 180 °C was considered in the calculation of AQEs. The radiative power of heat per unit area can be described by the Stefan–Boltzmann law[53],

$$Q = \sigma(T_r^4 - T_0^4), \quad (8)$$

where $T_r$ is the reaction temperature in K and $\sigma$ is Stefan–Boltzmann constant ($5.67 \times 10^{-8}$ W m$^{-2}$ K$^{-4}$). The maximal work done by the thermal energy was calculated by using the Carnot equation[54],

$$W_h = Q(1 - T_0/T_r), \quad (9)$$

where $T_0$ is the ambient temperature for performing the $CO_2$ reduction (299.15 K). Based on the above two equations, the real contribution of heat at 180 °C in $CO_2$ reduction was calculated to be 65.8 mW cm$^{-2}$. The AQE was calculated by:

$$
\begin{aligned}
AQE\ (\%) &= \frac{\text{Number of reacted electrons}}{\text{Number of incident photons}} \times 100\% \\
&= \frac{\text{Number of evolved CO molecules} \times 2}{\text{Number of incident photons}} \times 100\% \\
&= \frac{M \times N_A \times 2}{\frac{W \times A \times t \times \lambda}{h \times c}} \times 100\% \\
&= \frac{M \times N_A \times 2}{\frac{(W_l + W_h) \times A \times t \times \lambda}{h \times c}} \times 100\%,
\end{aligned}
\quad (10)
$$

where $M$ is the CO evolution rate (mol s$^{-1}$), $N_A$ is Avogadro constant, $W$ is the total energy input (W), $A$ is irradiation area (m$^2$), $t$ is the time of light illumination (s), $\lambda$ is the corresponding wavelength (m), $h$ is $6.62 \times 10^{-34}$ J s$^{-1}$, and $c$ is $3.0 \times 10^8$ m s$^{-1}$.

The electronic structure and free energy calculations were performed using spin-polarized DFT based on the Vienna ab initio simulation package[55]. PAW potentials were selected to describe ion core and valence electron interactions[56]. For the exchange-correlation functional, the generalized gradient approximation with the Perdew–Burke–Ernzerhof functional was adopted[57]. A kinetic energy cut-off of 400 eV was used with a plane-wave basis set. The integration of the Brillouin zone was conducted using a $1 \times 1 \times 1$ Monkhorst–Pack grid[58]. To explain the reaction activity from a kinetic aspect, the transition state structures and the reaction pathways were located using the climbing image nudged elastic band method[59]. The minimum energy path was optimized using a force-based conjugate-gradient method until the energy converged to $1.0 \times 10^{-4}$ eV atom$^{-1}$ and the maximum force was <0.05 eV Å$^{-1}$. Spin polarization and van der Waals (vDW) forces were considered in our current study using the vDW-DF method. The Bi(001) surface was obtained by cutting bulk Bi along the [001] direction. A $3 \times 3$ supercell with four layers was chosen as our model. During the geometry optimization, the atoms in the top three layers were allowed to relax, while the bottom layer was fixed. A vacuum layer of 15 Å was used along the $c$-direction normal to the surface to avoid periodic interactions.

To illustrate the $CO_2$ reduction reaction activity from a thermodynamic aspect, the free energy diagrams were estimated using:

$$\Delta G = \Delta E + \Delta ZPE - T\Delta S, \quad (11)$$

where $\Delta E$ is the total energy change based on the DFT calculations, ZPE and $S$ are the zero-point energy and entropy, respectively, and $T$ is the temperature (here,

298.15 K is selected). The free energy of $(H^+ + e^-)$ at standard conditions was assumed to be the energy of $1/2$ $H_2$[60].

## Data availability
The data that support the findings of this study are available from the corresponding author upon reasonable request.

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

## Acknowledgements

This work is financially supported by the National Natural Science Foundation of China (21972082 and 21673220). Part of the computational time is supported by the High Performance Computing Center of Jilin University, the High Performance Computing Center of Jilin Province, and network and computer center in Changchun Institute of Applied Chemistry, Chinese Academy of Sciences.

## Author contributions

Y.L. conceived and designed the experiments. D.H. performed the material synthesis, characterizations, and activity tests. Y.S. participated in material synthesis and activity tests. C.W. conducted the SEM and TEM measurements. Y.W. and Z.W. carried out DFT calculations. Y.L. wrote the manuscript. J.Z. and C.W. added to the discussion and contributed to the preparation of the manuscript. Y.L. and Y.W. supervised the work.

## Competing interests

The authors declare no competing interests.
