## [Peer Review File · Nature Communications]

REVIEWER COMMENTS

Reviewer #1 (Remarks to the Author):

Li et al present a manuscript on solar-assisted CO₂ reduction. Despite long-standing efforts for developing photocatalysts for efficient conversion of CO₂ into feedstock chemicals, the currently known materials generally have a low performance and have some additional challenges such as co-generation of O₂ and CO₂ reduction products. I really like the concept of spatially and temporally decoupling the oxidation and reduction half reactions of the process, but this is by no means a new idea. It has been previously applied to water splitting (e.g. Nature Energy 2019, 4, 786–795; Science 2014, 345, 1326–1330) which should be acknowledged here.

The authors claim that direct CO₂ reduction at semiconductors is not possible, but then include a list of examples from the literature in the SI that show exactly this reactivity. The paragraph describing this is misleading and should be rewritten. It is a common misconception that CO₂ reduction always has to go via the CO₂⁻ radical anion which has a redox potential of –1.9 V (not 1.9 V as it is written in the manuscript), but using catalysts can lower the required potential through stabilising or avoiding this high-energy intermediate. Similarly, the authors quote the high homolytic dissociation energy of H₂ to explain how it reacts. It is well known that H₂ in fact does not react via homolytic cleavage into radicals in the gas phase which then attack the surface, but via dissociative chemisorption to the surface, which has a much lower activation barrier. The authors also state the H₂ is expensive (not true, it is actually much cheaper than CO₂) and usually used in photocatalytic CO₂ reduction. To the best of my knowledge there is a lot more work using H₂O or sacrificial electron donors for this purpose than H₂.

Nevertheless, I like the idea of applying the decoupling concept to the half-reactions of CO₂ reduction, but I struggle to see the usefulness of the particular approach taken here. The authors are applying –1.2 V vs Ag/AgCl at pH 14 to generate a Bi-H intermediate which then reacts at >150°C with CO₂ when irradiated to give CO (all potentials should be given vs. RHE rather than vs Ag/AgCl in line with IUPAC recommendation to make the overpotentials more apparent). If I were to apply the same voltage (corresponding to 0.87 V overpotential for HER) to a cheap HER catalyst, e.g. Ni₂P and use the generated H₂ to hydrogenate CO₂, I would likely get much more CO and probably even more valuable chemicals such as MeOH. Do the authors observe any other CO₂ reduction products? It would also be possible to apply the same potential to a Bi-based electrode to directly reduce CO₂ with fairly decent performance and at room temperature (well-documented electrocatalytic activity e.g. J. Mater. Chem. A 2018, 6, 4714–4720 and many others), and with an OER anode separated by a membrane, CO₂ reduction and O₂ generation could also be spatially separated. The authors mention water splitting and overall CO₂ splitting several times throughout the manuscript and claim O₂ generation in the abstract, but in fact there is no water oxidation performed anywhere in the manuscript or SI, but only the reductive half-reaction. This must be rephrased in the manuscript.

From the data provided in the manuscript, it is difficult to work out exactly how efficient the presented system is. There is no chronoamperometry given, so it is not known how much charge is actually used to

generate the Bi-H_x intermediate; what is the faradaic yield of this process? The authors give an amount of Bi-H formed in the material and that only 25% of that is used to generate CO, but it is not clear how this was calculated. Can the presence of Bi-H be confirmed and quantified by Raman spectroscopy, ideally as a function of the applied bias/electrolysis time? Can the authors rule out that the increased activity of Bi-H is not only due to the increased presence of Bi(0) in the material? Is the ratio of Bi(0) and Bi(3) changed after photocatalysis?

The authors claim that the performance of their photocatalyst is much better than what is known from the literature. Is this really true? Table S1 is incomplete doesn't list the conditions under which the CO₂ reduction was performed, but I would assume that most of the examples were done at room temperature, whereas the present work was done at >150°C, and I would expect the performance to be drastically lower if the present system was operated at room temperature, so it is like comparing apples and oranges. In addition, the given examples actually use water as electron donor, whereas the present work uses Bi-H as sacrificial electron donor. The additional potential and efficiency losses of oxidising water would need to be factored in for a fair comparison. The observed quantum yield is also quite low, other (room temperature) processes achieve much better performances (e.g. ACS Appl. Energy Mater. 2020, 3, 5, 4509–4522). Temperatures and donors should be added to the table and it might be fairer to compare the present work with other examples of thermal CO₂ hydrogenation, which usually operates at >200°C, which isn't that far off and has much higher activities.

I am not 100% convinced by the mechanism. What exactly is being oxidised in the process? The fact that the non-hydrogenated catalyst still shows some activity is confusing. Where are the reducing equivalents coming from? What is being oxidised in this case? Do the authors have some experimental evidence to support the proposed mechanism or is this based entirely on calculations? What is being oxidised in the photoelectrochemical experiments?

Overall, I am not convinced this brings the quality and novelty needed for Nature Communications.

Reviewer #2 (Remarks to the Author):

Summary:

In the present work, Y. Li et al. study a hydrogen-stored Bi (Bi-H_x) system as an electron proton-transfer mediator for CO₂-to-CO conversion. They claimed that the stored hydrogen not only can react with CO₂ molecule to produce H₂O and CO but also it plays a role as a hole scavenger to promote the charge separation. The overall reaction is carried out under light at a certain temperature (180 °C), resulted in a high CO production yield of 283.3 μmol g⁻¹ h⁻¹. Using DFT calculations, they show that the energy barrier via a proton-assisted electron transfer pathway is much smaller on the Bi-H_x than the pristine Bi. However, they miss a deeper discussion on the material characterizations that are reflected in my comments below. This manuscript reports original results and a novel idea. Hence, I think this work can be published in Nature Communications after major revisions.

My comments:

1- The CO₂ reduction process is carried out under light at temperatures much higher than room temperature, which is used in the conventional photocatalytic CO₂ reduction. Therefore, I suggest the authors use the term “thermo-photocatalytic” (or “photo-thermocatalytic”, depending on the primary importance of a thermo- or photo-effect) in the title and other related parts of the manuscript.

2- (I) The XPS spectrum of the Bi-Hx was not fitted well (Fig. 2c): the FWHM of the oxidation state Bi³⁺ is much larger than that of metallic Bi. The same FWHM will cause a smaller blue shift relative to the pristine Bi. If they need to use different FWHM, it should be addressed in the manuscript. (II) Besides, the hydrogenation process resulted in a larger FWHM. Can it possibly suggest the presence of other oxidation states on the surface? (III) The authors should add deconvoluted carbon and oxygen spectra in the supporting information. These spectra can provide useful information about the probable existence of any additional oxidation states. (IV) As mentioned by authors (page 6 line 108-109), there are Bi₂O₃ at the very surface of the Bi-Hx. So, is it a kind of Bi-Hx@Bi₂O₃ core-shell system? How thick is this Bi₂O₃ “shell”?

3- Bi₂O₃ has a bandgap of about 2.5-2.7 eV. Therefore, it can show a peak at around 450-500 nm. So, it seems that the broad peak between 400-600 (in Fig. 2d) is due to the convolution of both LSPR of Bi-Hx and partial Bi₂O₃ absorption. If the nature of the peak is LSPR, the authors can provide several samples with different particle sizes (by using different electrodeposition times from 3 to 15 min). Then, if the nature of the absorption is LSPR, they would observe a shift in the UV-visible absorption.

4- SAED patterns should be recorded at the edge and basal plane of both Bi-Hx and Bi particles. It can show if there are any other local secondary phases or not.

5- The higher activity of the Bi-Hx/NF could be due to the formation of nanopores during the hydrogenation process, rather than, or not only just because of the proton-assisted electron transfer effect as claimed by the authors. Therefore, the authors should provide direct experimental evidence that the overall surface areas of both samples (Bi/NF and Bi-Hx/NF) are almost the same.

6- (I) For CO₂ reduction, the authors had applied heat, hence, there is additional input energy (ΔE), which should be added to the denominator of the AQE. In other words, the authors report the AQE values only based on photo-, but neglecting the thermo- contribution completely (Fig. 4f). (II) I will also suggest the authors to do the test under AM 1.5 solar light (only photo-catalysis, but not thermo-photo-catalysis). How much will the AQE be for photo-catalysis? The result shall be compared to other reported systems.

7- Current data did not provide convincing evidence that the hydrogen (in Bi-Hx lattice) can have bi-functional roles (page 12 line 231). Especially, for the claimed point 2, the authors should do the XPS after GC test for different times to prove that there is a continuous shift that can probably be assigned

to the consumption of H (i.e. H element belonging to the Bi-Hx phase).

8- The PEC measurements were not carried out in a stable condition. Degradation of the maximum photocurrent density is about 40% after several minutes (Fig. 5d)! The authors need to explain why? Or repeat the experiment.

9- As shown in XPS spectra, the very surface contains the Bi₂O₃ phase (quite dominant Bi³⁺ actually), which can change the calculated free energies (Fig. 6b). A serious concern is raised for the potential impact of this surface oxide layer on the whole CO₂ reduction process when it is in the vicinity of the Bi-Hx phase. The authors have claimed the energy profiles are not dramatically affected by the presence of Bi oxide (comparing Supplementary Fig. 2b and Main Fig. 6b), however, the computational results can depend on the content of oxide in the model. I think the authors must provide clear evidence, preferably experimental, showing that the surface oxide layer has not had a big impact on the proposed reaction pathway.

10- What will happen if the authors do the same experiment in CO₂ and H₂O atmosphere? Can the introduction of H₂O keep the concentration of hydrogen constant in the Bi-Hx system and provide better stability?

(Li-Chyong Chen)

Reviewer #3 (Remarks to the Author):

Dear Authors

find here my concerns.

First and foremost for any photocatalytic reaction needs to analyze the band gap of material which is not provided in manuscript.

In Fig 2d UV-Vis analysis doesn't show any significant difference between both catalyst then how both can behave differently prove it by providing additional analysis data like PL(Photo luminescence analysis) that can show the recombination of excitons. From Fig 2b cyclic voltametric analysis nickel foam is not much sensitive toward electrocatalytic reaction thus the use of Nickel foam must be explained.

From fig 2b It can also clearly observed the absorption/desorption peak of hydrogen that also questioned the stability of Bi-H x /NFi catalyst.

Fig2c XPS analysis shows the significant shift of binding energy though other analysis though there is no extra oxidation state of Bi is reported thus the change must be explained.

GC analysis presented the production of CO and hydrogen molecule in the system while photocatalytic reduction does not selective that clearly give the information about formation of other products or hydrocarbons, it require some additional analysis.

Comparative analysis data of ^{13}CO and CO is not provided in manuscript to prove the reliability of reaction mechanism needs to be rechecked.

Dissolution of gas in the water requires some extra outer forces to maintain the an equilibrium that is not explained in the manuscript need to be recheck and optimize the reactor conditions.

In DFT analysis favors the mechanism of reaction but the same time free energy value with respect to change in the transition, state shows reaction happened under the artificial condition, it was not natural that need to be explained.

Response to Reviewer 1's Comments:

1. Li et al present a manuscript on solar-assisted CO₂ reduction. Despite long-standing efforts for developing photocatalysts for efficient conversion of CO₂ into feedstock chemicals, the currently known materials generally have a low performance and have some additional challenges such as co-generation of O₂ and CO₂ reduction products. I really like the concept of spatially and temporally decoupling the oxidation and reduction half reactions of the process, but this is by no means a new idea. It has been previously applied to water splitting (e.g. Nature Energy 2019, 4, 786–795; Science 2014, 345, 1326-1330) which should be acknowledged here.

Response: The authors thank the reviewer for the thorough review. We thank the reviewer for liking “the concept of spatially and temporally decoupling the oxidation and reduction half reactions of the process”. We also appreciate the reviewers’ comments and suggestions that helped us to significantly improve the manuscript. We have addressed the reviewers’ comments point-wise in our response below.

The novelty of our manuscript lies in the development of a novel plasmonic Bi catalyst that spatially and temporally reduce CO₂ into CO efficiently by storing H species in Bi. Since CO₂ is a thermodynamically stable molecule, its reduction via the electrocatalytic or photocatalytic approach is significantly more challenging than the splitting of water, and confronted with many fundamental technical hurdles (*Adv. Sci.* 2017, 4, 1700194). Although the two steps process has been used in water splitting, the application of this concept in CO₂ conversion are still a challenge because the CO₂ reduction process with H₂O is much more complicated that contains the redox reactions between two different molecules of H₂O and CO₂. Furthermore, the material system, operation process and solved problems in the present study are quite different from the above references.

Action: In the revised version of manuscript, we have discussed this point in more detail on **Page 5** of the revised manuscript as shown in the following, and cited the relevant references in their proper places.

Similar with the natural photosynthesis, the decoupled approach for water splitting has been explored recently, in which water oxidation and proton reduction reactions were spatially separated.^{16,17} Compared with water splitting, CO₂ reduction is much more difficult because CO₂ is a thermodynamically stable molecule with linear structure¹. Furthermore, finding a single material to mimic natural photosynthesis for CO₂ conversion is more challenging because it should work as a redox shuttle to bridge the separated water splitting and CO₂ reduction reactions^{12,18}, which is more complicated than the only water splitting process.

2. The authors claim that direct CO₂ reduction at semiconductors is not possible, but then include a list of examples from the literature in the SI that show exactly this reactivity. The paragraph describing this is misleading and should be rewritten. It is a common misconception that CO₂ reduction always has to go via the CO²⁻ radical anion which has a redox potential of -1.9 V (not 1.9 V

as it is written in the manuscript), but using catalysts can lower the required potential through stabilising or avoiding this high-energy intermediate.

Response: We thank the reviewer for raising this question. This paragraph can indeed cause misleading to the reader as the reviewer indicated.

Action: We have re-written the whole paragraph at the beginning of the Introduction part that clarifies the importance of transferring proton-electron pairs to CO₂ as shown in the following.

Catalytic CO₂ reduction driven by solar light or renewable electricity to produce useful fuels is a promising strategy for solving energy and greenhouse effect issues¹. The CO₂ reduction activity is usually limited by the rate-limiting step of transferring an electron to a linear CO₂ molecule to form a bent CO₂^{•-} anion radical (equation 1)¹⁻³. After the rate-determining step, the intermediate CO₂^{•-} is subsequently reduced via the proton-assisted electron transfer approach, in which H₂ or H₂O is utilized as the proton source¹⁻³. In these CO₂ reduction processes, the types of the products are determined by the number of the transferred proton-electron pairs (equations 2-5)².

3. Similarly, the authors quote the high homolytic dissociation energy of H₂ to explain how it reacts. It is well known that H₂ in fact does not react via homolytic cleavage into radicals in the gas phase which then attack the surface, but via dissociative chemisorption to the surface, which has a much lower activation barrier.

Response: The authors agree, and appreciate the reviewer for catching this mistake.

Action: The following sentence describing the difficulty of H₂ dissociation in the initial submission was replaced by the following sentence in the revised manuscript on **Page 4**.

Furthermore, the H₂ dissociation process inevitably requires an energy input, whose value depends on the used catalysts.

4. The authors also state the H₂ is expensive (not true, it is actually much cheaper than CO₂) and usually used in photocatalytic CO₂ reduction. To the best of my knowledge there is a lot more work using H₂O or sacrificial electron donors for this purpose than H₂.

Response: We certainly agree with the reviewer that H₂ is not expensive. In the second paragraph of the Introduction part, the functions of two different proton sources (H₂ and H₂O) in CO₂ reduction were compared and discussed. In fact, we want to express that H₂ is more expensive than H₂O. However, we did not express this accurately in our original manuscript.

Action: To avoid any misleading, we have revised the corresponding sentences in the manuscript to make this point more clearly. The following sentences were added into the revised manuscript on **Page 4** as:

Compared with H₂O, the relatively high cost and explosive feature of H₂ may be two drawbacks for its application in CO₂ conversion.

5. Nevertheless, I like the idea of applying the decoupling concept to the half-reactions of CO₂ reduction, but I struggle to see the usefulness of the particular approach taken here. The authors are applying -1.2 V vs Ag/AgCl at pH 14 to generate a Bi-H intermediate which then reacts at >150°C with CO₂ when irradiated to give CO (all potentials should be given vs. RHE rather than vs Ag/AgCl in line with IUPAC recommendation to make the overpotentials more apparent).

Response: We thank the reviewer for liking the two-step concept for CO₂ reduction in our present work. Our response on this question is divided into two separated parts on the high reaction temperature and electrochemical potential. As discussed below, both the Bi sample that can work under at high temperature, and the electrochemical process for H storage have their own unique advantages in CO₂ reduction.

Response (I): About the reaction temperature

The present Bi nanosheets are a special metal catalyst that has the surface plasmon resonance (SPR) property. We acknowledge the high temperature used in our study, although we do not utilize precious materials, sacrificial organics or cocatalysts. To the best of our knowledge, all the reported plasmonic photocatalysts work only when the reaction temperature is relatively high (>150 °C). For example, the reaction temperatures for photocatalytic CO₂ conversion are 200-350 °C for plasmonic Au, and Rh (*Nat. Commun.* 2017, 8, 14542), 400 °C for plasmonic Pt-Au (*ACS Appl. Mater. Interfaces* 2018, 10, 408). Based on our understanding, plasmonic photocatalysis driven by photoinduced hot-electrons is a type of catalysis that between thermal catalysis and photocatalysis. Therefore, it is reasonable that heat input is needed for activating the adsorbed molecules. Furthermore, most of the plasmonic photocatalysts are noble metal nanoparticles, such as Ag, Au, and Rh. **While the present Bi is the first example of hot carrier driven highly selective CO₂-to-CO conversion using a plasmonic metal without combining semiconductors that entirely relies on nonprecious, earth-abundant materials.** The discussion on this point was added in our revised manuscript, which could bring novelty of our manuscript with clarity.

The field of heterogeneous photocatalysis has almost exclusively focused on semiconductor photocatalysts (*Nat. Mater.*, 2015, 14, 567). Compared with the semiconductor-based photocatalysis that has dominated the research area since 1972, plasmonic photocatalysis just started from the year 2011 (*Nat. Chem.*, 2011, 3, 467). Before this report, plasmonic metals are mainly used as thermal catalyst or as light absorber by combining with semiconductor. Therefore, plasmonic metallic nanostructures represent a new family of photocatalysts, and only a few reports focus on CO₂ reduction (*Adv. Mater.* 2020, 2000086). In the thermochemical CO₂-to-CO conversion process on metal catalysts, it is usually required a very high temperature of 550–750 °C (*ACS Energy Lett.* 2018, 3, 1938). Furthermore, the applications of photocatalytic systems based on plasmonic metals loaded semiconductors are significantly hindered by the low hot charge transfer efficiency from metal to catalytically active sites on the semiconductor, which is attributed to the built-in Schottky barrier at the semiconductor-metal interface (*J. Am. Chem. Soc.* **2017**, 139, 17964).

Although conventional semiconductor photocatalysts offer a promising route to room temperature reactions, in general they exhibit lower reaction rates at higher temperatures due to the relatively low Debye temperatures of the semiconductors (*Nat. Mater.* 2012, 11, 1044; *Nat. Commun.* 2017, 8, 14542). Therefore, in practical applications, if we want to increase the yield of products, the efficiencies for general photocatalysts can be significantly reduced by increasing the light intensity, which might be a potential factor that can hinder use of semiconductor photocatalysts because high-intensity light, especially IR light, will inevitably increase the temperature of the gas-solid reaction system (*Nat. Commun.* 2017, 8, 14542).

The plasmonic Bi nanoparticles in the present manuscript characterized by excellent absorption of visible light through the creation of resonant surface plasmons, can utilize concurrently thermal energy and a photon flux to drive catalytic reactions at moderately high temperature of 180 °C, which is much lower than those associated with conventional thermal processes. In fact, the heat input can be provided by the infrared light of sunlight (accounting for ~50% of the solar energy), which implies that the plasmonic Bi photocatalysts can potentially use the entire solar spectrum for driving catalytic reactions by employing a high-temperature solar reactor driven by concentrated solar radiation. In this respect, the ability that plasmonic photocatalysts can work at high temperature have their own unique advantages in CO₂ reduction.

Action (I): To address this question from the reviewer, the following sentences were added into the revised manuscript on **Pages 9-10** and **Page 11-12** as:

Compared with the semiconductor-based photocatalysis that has received the most attention for several decades, plasmonic photocatalysis just started from the year 2011²⁸. Therefore, plasmonic metallic nanostructures represent a new family of photocatalysts, and even a few reports focus on CO₂ reduction²⁹. In plasmonic photocatalysis, a relatively high reaction temperature (>150 °C) is generally required for activating the adsorbed molecules²⁹⁻³¹. Therefore, it is reasonable that the photocatalytic CO₂ reduction on the plasmonic Bi should be triggered with heat input. Although most plasmonic metals used as light absorber in metal-semiconductor systems can induce photocatalytic reactions at room temperature, the low charge transfer efficiency between metals and semiconductors is the key to limiting their applications³². Furthermore, most of the plasmonic photocatalysts for CO₂ reductions are based on noble metal nanoparticles, such as Ag, Au, and Rh^{30,31}. While the present Bi is the first plasmonic photocatalyst for CO₂ conversion by using non-precious metals without combining semiconductors.

Considering that the heat input can be provided by the infrared light of sunlight (accounting for ~50% of the solar energy), thus the plasmonic Bi photocatalysts can potentially use the entire solar spectrum for driving catalytic reactions by employing a high-temperature solar reactor driven by concentrated solar radiation. Although conventional semiconductor photocatalysts can work at room temperature, their activities always decreased with increased temperatures due to the relatively low Debye temperatures of the semiconductors^{30,35}. Therefore, in practical applications, the heat effect

from concentrated sunlight might be one of the hindrances for efficient solar energy conversion in gas-solid reaction system by using semiconductor photocatalysts³⁰. In this respect, the ability of the plasmonic Bi photocatalyst that can work at 180 °C might have its own advantage for CO₂ reduction in gas-solid reactions.

Response (II): About the electrochemical problems

According to the reviewer's suggestion, all the potentials in the revised manuscript were given by RHE. The corresponding formula was included in Methods part in the revised manuscript. The cyclic voltammograms of the as-prepared Bi/NF was modified accordingly as shown in the **Fig. R1** below. Electrochemical hydrogen storage in Bi was performed at a reduction potential of -0.18 V versus RHE (vs. RHE), which is not so high.

The unique advantage of the electrochemical process used in the present work is to convert H⁺ in water to active H bounded on the surface of the Bi catalyst that can perform CO₂ reduction with a much higher activity than H₂. Traditionally, H₂, which can be produced by electrocatalytic water splitting, steam reforming of natural gas or et al., was used for CO₂ reduction on metals. In these CO₂ reduction processes, the molecular hydrogen should be decomposed into atomic hydrogen first, which always requires an energy input. Compared with H₂ gas, atomic H atoms preformed on the surface not only improve photoinduced charge separation, but also lower the CO₂ reduction barriers as discussed in Fig. 5 and Fig. 6 in the revised manuscript. Therefore, the electrochemical hydrogen storage process provided an effective approach to produce reactive H for CO₂ reduction by transforming protons in water, which is much more active than traditional H₂ (Fig. 4e in the revised manuscript). This is one advantage for introducing electrochemical process in the catalytic circle. Furthermore, the present two-step approach provide an effective model for reducing CO₂ by using cheap and abundant H₂O. Given the wide diversity of known H-intercalating materials with a wide range of H-affinities, the concept established here provides the basis of segregating bond activation and charge separation processes across a wide range of key energy conversion reactions.

Fig. R1 Cyclic voltammograms of the as-prepared Bi/NF and pristine NF in 1 M KOH solution. Scan rate: 50 mV/s. (This figure has been updated in the revised manuscript as Fig. 2b.)

Action (II): We have provided additional discussion based on the reviewer's comments. The advantages of the electrochemical hydrogen storage process were added in the revised manuscript as shown in the response on the following comment (5), and the discussion in the DFT calculations in Fig. 6 of the revised manuscript. The following sentences were added into the revised manuscript on **Pages 14-15, and 23**. Furthermore, all the potentials were given by RHE in the revised manuscript.

For the CO₂ reduction process on Bi/NF with H₂ gas, the photocatalytic interface must facilitate two sequence steps (1) the adsorption and decomposition of the H₂ to generate atomic H bound on the surface, and (2) the transfer of these adsorbed H atoms into electron-proton pairs to reduce CO₂ (Fig. 4g). However, the two coupled steps (1 and 2) in both time and space preventing independent optimization of each, and the competitive adsorption of H₂ may impede CO₂ activation at the active sites³⁶. Moreover, the decomposition of the H₂ in step (1) will inevitably require a certain energy input. By separating the above two steps (1 and 2), the present approach for CO₂ conversion addresses the incompatibilities of functions (1) and (2), and bypasses the H₂ decomposition process in traditional CO₂ reduction process with H₂ (Fig. 4h). As a result, the improved CO₂ reduction activity on Bi-H_x/NF can be attributed to the atomic H atoms preformed on the surface, which not only improve photoinduced charge separation, but also lower the CO₂ reduction barriers. These two positive effects will be closely discussed in the following Fig. 5 and Fig. 6, respectively.

Considering the existence of large variety of H-intercalating materials, the concept developed here may have broad applications for developing efficient and multifunctional catalysts for CO₂ conversion by mimicking photosynthesis.

5. If I were to apply the same voltage (corresponding to 0.87 V overpotential for HER) to a cheap HER catalyst, e.g. Ni₂P and use the generated H₂ to hydrogenate CO₂, I would likely get much more CO and probably even more valuable chemicals such as MeOH.

Response: We agree that electrocatalytic H₂O splitting is an appropriate approach for producing H₂ by using some nonprecious metal based electrocatalysts. However, the electrocatalysts, such as Ni₂P, cannot catalyse the further hydrogenation reaction of CO₂. The fascinating point of the Bi catalyst is the bifunctional properties that can not only induce electrocatalytic H₂O splitting and H storage, but also drive the CO₂ reduction reactions under light irradiation. More importantly, the surface-bound H atoms is a more efficient proton source than H₂ for CO₂ reduction process in the same photochemical reaction conditions, which is another main point of this manuscript.

It is well known that H₂ is traditionally used as the proton source for photochemical CO₂ conversion via the proton-assisted electron transfer approach. In this process, photochemical CO₂ conversion relies on the transfer efficient of electron-proton pairs from the H₂ molecules under light irradiation, and the photochemical interface must facilitate two sequence steps (1) the decomposition of the H₂ to generate surface-bound H atoms, and (2) the transfer of these adsorbed H atoms into electron-proton pairs to reduce CO₂. However, the two coupled steps (1 and 2) in both time and space

preventing independent optimization of each, and competitive adsorption of H_2 may impede CO_2 activation at the active sites. Moreover, the decomposition of the H_2 in step (1) will inevitably require a certain energy input.

In the present manuscript, we establish a novel strategy for CO_2 reduction by exploiting an electron-proton-transfer mediator ($Bi-H_x$) with recyclability, which was generated from electrochemical H_2O splitting of Bi electrode. By segregating the above two steps (1 and 2), the strategy in our manuscript addresses the incompatibilities of functions (1) and (2), and avoids the H_2 activation process in traditional CO_2 reduction process using H_2 as proton source. In order to show that the current approach using surface-bound H atoms as proton source has advantages over the traditional H_2 , the controlled experiments for CO_2 reduction were carried out on the same Bi material under the similar experimental conditions. As shown in Fig. 4e, the photochemical CO production activity using surface-bound H atoms on $Bi-H_x$ as proton source is 9 times that on pristine Bi using H_2 molecules as the proton source (Fig. 4e) because the preformed H atoms on the $Bi-H_x$ surface can improve photoinduced charge separation and lower the CO_2 reduction barriers as discussed in Fig. 5 and Fig. 6 in our revised manuscript.

Fig. R2 Schematic diagram of CO_2 reduction processes using H_2 as proton source on **a** Bi/NF and **b** preformed H species as proton source on $Bi-H_x$ /NF. (This figure has been added into the revised manuscript as new Fig. 4g and h.)

Action: To address this question from the reviewer, schematic representations of CO_2 reduction by (a) a traditional H_2 molecules in one step process and (b) $Bi-H_x$ in the two-step approach were added in the revised manuscripts as shown in Fig. 4 g and h. Except on the discussions on Fig. 5 and Fig. 6 in our original manuscript, the following text was added into the revised manuscript on **Page 14-15** as:

For the CO_2 reduction process on Bi/NF with H_2 gas, the photocatalytic interface must facilitate two sequence steps (1) the adsorption and decomposition of the H_2 to generate atomic H bound on the surface, and (2) the transfer of these adsorbed H atoms into electron-proton pairs to reduce CO_2 (Fig. 4g). However, the two coupled steps (1 and 2) in both time and space preventing independent

optimization of each, and the competitive adsorption of H₂ may impede CO₂ activation at the active sites³⁶. Moreover, the decomposition of the H₂ in step (1) will inevitably require a certain energy input. By separating the above two steps (1 and 2), the present approach for CO₂ conversion addresses the incompatibilities of functions (1) and (2), and bypasses the H₂ decomposition process in traditional CO₂ reduction process with H₂ (Fig. 4h). As a result, the improved CO₂ reduction activity on Bi-H_x/NF can be attributed to the atomic H atoms preformed on the surface, which not only improve photoinduced charge separation, but also lower the CO₂ reduction barriers. These two positive effects will be closely discussed in the following Fig. 5 and Fig. 6, respectively.

6. Do the authors observe any other CO₂ reduction products?

Response: Other products such as CH₄, and CH₃OH were not observed by mass spectrum from GC-MS and GC analysis.

Action: The corresponding description on this point has been added into the revised manuscript on **Pages 10, 12, and 25** as:

Other possible products (such as CH₄, and CH₃OH) were not detected, confirming that high selectivity for CO₂ reduction was achieved on Bi-H_x/NF.

As shown in Fig. 4c, except for the peaks belonging to CO₂ and CO, no other peaks were observed in Fig. 4c, further confirming the high selectivity for CO₂ reduction on Bi-H_x/NF. This result is consistent with that of gas chromatography.

For detecting CH₃OH, the gaseous products from the reactor were analysed by an offline GC (Fuli Corp., China) equipped with a flame ionized detector (FID). The equipped column was TDX-01.

7. It would also be possible to apply the same potential to a Bi-based electrode to directly reduce CO₂ with fairly decent performance and at room temperature (well-documented electrocatalytic activity e.g. *J. Mater. Chem. A* 2018, 6, 4714-4720 and many others), and with an OER anode separated by a membrane, CO₂ reduction and O₂ generation could also be spatially separated.

Response: Although the electrocatalytic approach can be operated at room temperature, some drawbacks also exist in this method. As we know that formate is mainly produced on the Bi-based electrodes based on the references (*J. Mater. Chem. A* 2018, 6, 4714; *Nat. Commun.* 2018, 9, 1320; *Nat. Commun.* 2019, 10, 2807). However, CO₂ reduction to liquid fuels by electrochemical approach is currently challenged by low product concentrations, as well as their mixture with traditional liquid electrolytes, such as KHCO₃ solution (*Nat. Commun.* 2020, 11, 3633). Therefore, the product cannot be directly used without further purification. The low concentration nature of the liquid fuel makes the purification process more difficult.

Based on the above references on Bi electrocatalyst, the major product in CO₂ reduction is formate (HCOO⁻). Different from acidic form, formate has no obvious usage (*Angew. Chem. Int. Ed.* 2018,

57, 2943). In sharp contrast, CO gas is a critical feedstock for directly synthesizing a variety of chemicals in industry (*Angew. Chem. Int. Ed.* 2018, 57, 2943). Therefore, the CO₂-to-CO conversion with high selectivity achieved on the present Bi catalyst has great potential in practical applications. Furthermore, the electrocatalytic approach displays poor solubility for gaseous substrates and limits the rate of the substrate bond activation. Therefore, it is difficult to perform direct CO₂ reduction in electrocatalytic approach (*J. Am. Chem. Soc.* 2019, 141, 28, 11115). Finally, the membrane used in electrocatalysis is expensive and the membrane degradation tends to be occurred at low current densities (*Science* 2014, 345, 1326-1330). It is lucky that the above three drawbacks do not exist in the present Bi system for CO₂ reduction. However, it should be keep in mind that there remains a noticeable absence of a consummate solution that combines all the ideal properties together in one material.

Action: We have provided additional discussion based on the reviewer's comments. The following sentences were added into the revised manuscript on **Page 22** as:

Although electrocatalytic reduction of CO₂ into formate (HCOO⁻) on Bi electrode can reach a much higher FE efficiency than the electrochemical H₂O splitting on the present Bi/NF, much higher overpotentials between -0.67 and -0.87 V (vs. RHE) were used in previous reports⁴³⁻⁴⁵. Furthermore, the direct reduction of CO₂ by the electrochemical approach is not a perfect technology because it still suffers from low product concentrations in the mixture of traditional liquid electrolytes⁴⁶. As a result, the product cannot be directly used without further purification. However, the purification of the low concentration liquid fuel in electrolytes not only compromises energy efficiency, but also is a technical challenge⁴⁶.

It is well known that different from acidic form, formate has no obvious usage⁴⁷. In contrast, CO gas is a critical feedstock for directly synthesizing a variety of chemicals in industry⁴⁷. Therefore, compared with electrochemical CO₂ reduction on Bi, the CO₂-to-CO conversion achieved on the present Bi catalyst has great potential in practical applications. Furthermore, the electrocatalytic approach displays poor solubility for gaseous substrates and limits the rate of the substrate bond activation. Therefore, it is difficult to perform direct CO₂ reduction in electrocatalytic approach³⁶. In addition, although the gas product and O₂ generation in electrocatalytic CO₂ conversion could also be spatially separated by a membrane, the used membrane is expensive and tends to be degraded at low current densities¹⁷. However, the above three drawbacks do not exist in the present catalytic system for CO₂ reduction. Based on our understanding, there is a noticeable absence of a consummate material that can solve all of these challenges in CO₂ reduction together.

8. The authors mention water splitting and overall CO₂ splitting several times throughout the manuscript and claim O₂ generation in the abstract, but in fact there is no water oxidation performed anywhere in the manuscript or SI, but only the reductive half-reaction. This must be rephrased in the manuscript.

Response: Thanks for your suggestion. The produced H₂ and O₂ products in electrochemical water splitting were identified by gas chromatography (Agilent 7890B) and Neofox-GT oxygen probe (Ocean opticals), respectively.

Fig. R3 H₂ and O₂ evolution curves on the Bi/NF electrode by water splitting at 0.18 V (vs. RHE). (This figure has been added into the revised manuscript as new Fig. 2c.)

Action: The corresponding O₂ production has been provided in **Fig. R3**. The following sentences were added into the revised manuscript on **Page 6-7** as:

In this process, the H₂ and O₂ evolutions by water splitting are shown in Fig. 2c. The obtained products exhibit a nearly linear rise in time from 0 to 10 min. It is worth noting that the molar ratio of H₂/O₂ is about 0.52, which is much lower than the stoichiometric ratio of 2. This result suggests that a part of the reduced H species was stored in Bi electrode without desorption, which is consistent with the CV curve of Bi/NF in Fig. 2b. The number of hydrogen atoms (M_H) stored in Bi-H_x/NF was estimated to be 13.52 µmol based on the amounts of the produced H₂ (M_{H2}) and O₂ (M_{O2}) in Fig. 2c (M_H = 4M_{O2} - 2M_{H2}).

9. From the data provided in the manuscript, it is difficult to work out exactly how efficient the presented system is. There is no chronoamperometry given, so it is not known how much charge is actually used to generate the Bi-H_x intermediate; what is the faradaic yield of this process?

Response: Faradaic efficiency (FE) of the Bi electrocatalyst for H₂O splitting was calculated to be 20.9% based on the O₂ evolution. The low FE of the Bi electrocatalyst (20.9%) in the decomposition of H₂O is a drawback for the present Bi catalyst. Although Bi-based electrode can directly reduce CO₂ into formate (HCOO⁻) with a much higher efficiency than the present Bi-H_x/NF (*J. Mater. Chem. A* 2018, 6, 4714; *Nat. Commun.* 2018, 9, 1320; *Nat. Commun.* 2019, 10, 2807), much higher

overpotentials between -0.67 and -0.87 V (vs. RHE) were used. Furthermore, the present manuscript does not aim to develop high efficient electrocatalyst, but rather demonstrate a novel two-step reaction model mediated by plasmonic Bi-H_x for CO₂ conversion by mimicking natural photosynthesis, which not only improves the CO₂ conversion activity but also avoids O₂ separation from CO product. However, improving the electrochemical performance of Bi is very important for reducing the overall energy input in the present two-step CO₂ reduction process, which is the subject of ongoing investigations in our laboratory.

Action: We tested faradaic yield for H₂O splitting using Bi electrode. The data are included in revised version. The following sentence was added into the revised manuscript on **Page 7 and Page 21-22** as:

Moreover, Faradaic efficiency (FE) of the Bi electrocatalyst for H₂O splitting was calculated to be 20.9% based on the O₂ evolution according to equation (7) in Methods part.

It should be indicated that the low FE of the Bi catalyst (20.9%) in the decomposition of H₂O is a drawback for it. However, the present manuscript does not aim to develop high efficiency electrocatalysts, but rather demonstrates a novel two-step reaction model mediated by plasmonic Bi-H_x for CO₂ conversion by mimicking photosynthesis. Improving the electrochemical performance of Bi is important for reducing the overall energy input in the present two-step CO₂ reduction process, which is the subject of ongoing investigations in our laboratory.

Although electrocatalytic reduction of CO₂ into formate (HCOO⁻) on Bi electrode can reach a much higher FE efficiency than the electrochemical H₂O splitting on the present Bi/NF, much higher overpotentials between -0.67 and -0.87 V (vs. RHE) were used in previous reports⁴³⁻⁴⁵. Furthermore, the direct reduction of CO₂ by the electrochemical approach is not a perfect technology because it still suffers from low product concentrations in the mixture of traditional liquid electrolytes⁴⁶. As a result, the product cannot be directly used without further purification. However, the purification of the low concentration liquid fuel in electrolytes not only compromises energy efficiency, but also is a technical challenge⁴⁶.

10. The authors give an amount of Bi-H formed in the material and that only 25% of that is used to generate CO, but it is not clear how this was calculated. Can the presence of Bi-H be confirmed and quantified by Raman spectroscopy, ideally as a function of the applied bias/electrolysis time?

Response: The 25% H used to generate CO is a misunderstanding, which was caused by our unclear description. The fact is that we compared the CO₂ activity of Bi-H_x with pristine Bi (using H₂ as the proton source) in the same experimental conditions. We want to express that the total amount of protons stored in Bi-H_x is 25% of that added in Bi reaction system (calculated from the added amount of H₂ molecules).

Action: We have revised the corresponding sentences in the revised manuscript on **Pages 13** to make this point more clearly:

It is well known that H_2 is traditionally used as the proton source for photochemical CO_2 conversion via the proton-assisted electron transfer approach³. In order to show that the current approach using surface-bound H atoms as proton source has advantages over the traditional H_2 , the controlled experiment for CO_2 reduction was carried out on Bi/NF material in dissociative H_2 gas (0.5 mL). As shown in Fig. 4e, although the number of hydrogen atoms stored in the Bi- H_x /NF systems (13.52 μ mol) is only about 25% of that from H_2 in Bi/NF reaction system, the CO_2 reduction activity of Bi- H_x /NF reaches 9 times that of Bi/NF.

11. Can the authors rule out that the increased activity of Bi-H is not only due to the increased presence of Bi(0) in the material? Is the ratio of Bi(0) and Bi(3) changed after photocatalysis?

Response (I): We thank the reviewer for this constructive comment. In order to exclude that the increased activity of Bi- H_x is not caused by the increased amount of Bi(0) in the material, the Bi- H_x sample contains no surface Bi(0) was prepared by treating the sample in air at 80 °C for 0.5 h. As proved by the XPS in **Fig. R4** below, there was no Bi(0) existed on the surface of the sample after the heat treatment. Then, the controlled experiment for CO_2 reduction was carried out on the Bi- H_x sample after heat treatment. By comparing the activity in **Fig. R5** below with that in Fig 4d in the revised manuscript, no catalytic performance degradation was found when Bi(0) was replaced by Bi(3), indicating that that the improved activity of Bi- H_x /NF as shown in Fig. 4e in the revised manuscript cannot be attributed to the increased amount of surface Bi⁰, which is consistent with the DFT calculations in Fig. 6.

Fig. R4 High-resolution XPS spectra of Bi 4f of Bi- H_x /NF sample treated in air at 80 °C for 0.5 h.
(This figure has been added into the revised Supplementary Information as new Fig. S6.)

Fig. R5 Reduction of CO₂ of Bi-H_x/NF samples after heat treatment. (This figure has been added into the revised manuscript as new Fig. 4f.)

Action: The following discussions were added into the revised manuscript on **Page 13-14** as:

As shown in the XPS spectra in Fig. 2d, the amount of Bi⁰ species on the surface of Bi-H_x/NF is much larger than that of Bi/NF. In order to exclude that the enhanced activity of Bi-H_x/NF in Fig 4e is not caused by the increased content of Bi⁰, the Bi⁰ on the surface of Bi-H_x/NF was oxidized into Bi³⁺ by treating the sample in air at 80 °C for 0.5 h. The absence of Bi⁰ species on the surface of Bi-H_x/NF after the treatment can be proved by XPS measurement (Supplementary Fig. 6). In the following, the controlled experiment for CO₂ reduction was carried out on the sample after heat treatment. Compared with the activity of Bi-H_x/NF in Fig. 4d, no performance degradation on CO production was found on Bi-H_x/NF when the surface Bi⁰ species was replaced by Bi³⁺ (Fig. 4f), indicating that the improved activity of Bi-H_x/NF in Fig. 4e cannot be attributed to the increased amount of surface Bi⁰.

Response (II): According to the suggestion of the reviewer, the XPS of Bi-H_x/NF was tested after 27 h thermo-photocatalytic CO₂ reduction. The result indicated that the content of Bi(0) after one reaction circle shows no obvious change.

Fig. R6 XPS spectra of Bi-H_x/NF after different thermo-photocatalytic reaction times. (This figure has been added into the revised Supplementary Information as new Fig. 11.)

Action: The XPS of Bi-H_x/NF after one reaction circle was tested as shown in **Fig. R6**. We have provided corresponding detailed discussion in the revised manuscript on **Pages 19** as:

In addition, based on the XPS analysis of Bi-H_x/NF, no obvious change on the ratio of Bi⁰ and Bi³⁺ is observed after 27 h reaction, indicating the good stability of the catalyst under thermo-photocatalytic reaction conditions.

12. The authors claim that the performance of their photocatalyst is much better than what is known from the literature. Is this really true? Table S1 is incomplete doesn't list the conditions under which the CO₂ reduction was performed, but I would assume that most of the examples were done at room temperature, whereas the present work was done at >150°C, and I would expect the performance to be drastically lower if the present system was operated at room temperature, so it is like comparing apples and oranges. In addition, the given examples actually use water as electron donor, whereas the present work uses Bi-H as sacrificial electron donor. The additional potential and efficiency losses of oxidising water would need to be factored in for a fair comparison. The observed quantum yield is also quite low, other (room temperature) processes achieve much better performances (e.g. ACS Appl. Energy Mater. 2020, 3, 5, 4509–4522). Temperatures and donors should be added to the table and it might be fairer to compare the present work with other examples of thermal CO₂ hydrogenation, which usually operates at >200°C, which isn't that far off and has much higher activities.

Response: We agree that it is very difficult to compare our results with the literatures. This is not because of our experimental conditions, but rather the significant diversity of utilized materials and reported experimental conditions for photocatalytic CO₂ conversion.

Improving quantum yield is the Holy Grail in all photocatalytic processes, and in particular for CO₂ conversion. However, the present manuscript should not emphasize CO₂ conversion efficiency after carefully considering the reviewer' comments, but rather report the following two important findings: First, our study demonstrates a novel two-step reaction model mediated by plasmonic Bi-H_x for CO₂ conversion by mimicking natural photosynthesis, which not only improves the CO₂ conversion activity but also avoids O₂ separation from CO product. Second, we also demonstrates the first example of hot carrier driven highly selective CO₂-to-CO conversion using bare plasmonic metal nanoparticles that entirely relies on nonprecious, earth-abundant materials. To improve the quality of this manuscript, we have provided an additional discussion about this advantage of our study in the revised manuscript.

We believe it is important to compare the apparent quantum efficiency of our system with other plasmonic systems that have reported photocatalytic CO₂ conversion. Even though there are dozens of reports at room temperature, our measured CO formation rate of 283.8 μmol·g⁻¹·h⁻¹ on Bi-H_x is even higher than many previous reports of photocatalytic CO₂ conversion (*ACS Catal.* 2016, 6, 7485–7527). Even though there are lots of reports on plasmonic metals for photocatalytic CO₂ reduction, these plasmonic metals were mainly coupled with semiconductor photocatalysts. To date, only several reports for photocatalytic CO₂ conversion were focused on plasmonic systems without coupling semiconductor, and all of them were based on noble-metals such as Au, Au-Ag, and Rh (*Nat. Commun.* 2017, 8, 14542; *ACS Appl. Mater. Interfaces* 2018, 10, 408). To objectively understand the quantum efficiency of the plasmonic Bi catalysts, activities of thermal CO₂ hydrogenation on metals, photocatalytic CO₂ reduction based on plasmonic metal (Au, Ag, and Rh), and plasmonic metal coupled with semiconductor systems were discussed systematically based on reaction temperature, and the used light intensity in the revised manuscript.

Action: In the revised version of manuscript, we have discussed and compared our efficiencies with literature reports on photocatalytic CO₂ conversion as well as similar plasmonic systems for CO₂ conversion. The Refs. (*Nat. Commun.* 2017, 8, 14542; *ACS Appl. Mater. Interfaces* 2018, 10, 408; *ACS Appl. Energy Mater.* 2020, 3, 5, 4509–4522 and *Nat. Commun.* 2017, 8, 27; *Adv. Mater.* 2020, 2000086 and *et al.*) were added in the revised manuscript. The following sentence was added into the revised manuscript on **Pages 10 and 15-16** as:

Although most plasmonic metals used as light absorber in metal-semiconductor systems can induce photocatalytic reactions at room temperature, the low charge transfer efficiency between metals and semiconductors is the key to limiting their applications³². Furthermore, most of the plasmonic photocatalysts for CO₂ reductions are based on noble metal nanoparticles, such as Ag, Au, and Rh^{30,31}. While the present Bi is the first plasmonic photocatalyst for CO₂ conversion by using non-precious metals without combining semiconductors.

Obviously, the plasmonic photocatalyst of Bi-H_x/NF is a type of catalyst that can combine light and thermal energy together for exciting CO₂ reductions. To objectively understand the quantum efficiency of the plasmonic Bi catalysts, examples of thermal CO₂ hydrogenation on metals, photocatalytic CO₂ reduction based on plasmonic metals, and plasmonic metal coupled with semiconductor systems are compared. Although a temperature of 180 °C is needed in the present Bi-H_x/NF system, this temperature is much lower than the thermochemical CO₂ conversion process on metal catalysts (550–750 °C)³⁷. Recently, plasmonic photocatalysis based on noble metal nanostructures was reported for CO₂ reduction with a much higher activity than the present Bi catalyst^{30,31}. It should be indicated that except for a high reaction temperature, the high activities on the noble metals were obtained under intense light illuminations (above 1 W•cm⁻²). However, the present Bi catalyst can induce CO₂-to-CO conversion with a relatively high CO evolution rate of 283.8 μmol•g⁻¹•h⁻¹ under low-intensity LED light illumination (108.6 mW•cm⁻², around solar intensity). Moreover, in comparison to semiconductor systems coupled with plasmonic metals that can proceed at room temperature, the present catalytic circle for CO₂ reduction shows a much higher photocatalytic activity although some semiconductors with significantly high efficiencies were reported^{1,4,29,38,39}. More details on the photocatalytic CO₂ reduction efficiencies on semiconductor systems can be found in the recent review articles^{1,4}. Rather than the catalytic efficiency, the main point of the present manuscript is to demonstrate a novel two-step reaction model for CO₂ conversion by using a plasmonic Bi metal that entirely relies on nonprecious materials.

13. I am not 100% convinced by the mechanism. What exactly is being oxidized in the process? The fact that the non-hydrogenated catalyst still shows some activity is confusing. Where are the reducing equivalents coming from? What is being oxidized in this case? Do the authors have some experimental evidence to support the proposed mechanism or is this based entirely on calculations? What is being oxidised in the photoelectrochemical experiments?

Response: This is a very reasonable concern. Our response on this question is divided into two separated parts as shown in the following.

Response (I): The Bi catalyst without hydrogen storage showed CO₂ reduction activity because H₂ molecules were used as reducing agent and as the proton source. In this process, H₂ was oxidized under light illumination. We are very sorry for the unclear description in our original manuscript, which indeed cause misleading to the reader as the reviewer indicated. The authors appreciate the reviewer for catching this mistake.

Fig. R7 Comparison on the thermo-photocatalytic CO production activities between Bi/NF using H₂ as proton source and Bi-H_x/NF using the stored H as proton source. (This figure has been undated in the revised manuscript as Fig. 4e.)

Action (I): We have replaced that statement with a clarified version in the main text. Please review **Page 13** of the revised manuscript for our response as shown in the following. For clarity, the descriptions in Fig. 4e were modified correspondingly as shown in the **Fig. R7** above.

It is well known that H₂ is traditionally used as the proton source for photochemical CO₂ conversion via the proton-assisted electron transfer approach³. In order to show that the current approach using surface-bound H atoms as proton source has advantages over the traditional H₂, the controlled experiment for CO₂ reduction was carried out on Bi/NF material in dissociative H₂ gas (0.5 mL). As shown in Fig. 4e, although the number of hydrogen atoms stored in the Bi-H_x/NF systems (13.52 μmol) is only about 25% of that from H₂ in Bi/NF reaction system, the CO₂ reduction activity of Bi-H_x/NF reaches 9 times that of Bi/NF.

Response (II): In order to experimentally prove that the thermal-photocatalytic CO₂ reduction on Bi-H_x was performed by the proton-assisted electron transfer approach, the following experiments were designed. First, CO₂ reduction on Bi-H_x/NF was performed at 180 °C without light illumination by using H₂O (200 μL) as proton source. The negligible CO production activity as shown in **Fig. R8** below demonstrates that photoinduced charges are the primary factor for CO₂ reduction on Bi-H_x/NF. Second, as shown in **Fig. R9** below, the formation of no CO on the bare Bi without storing H justifies the hypothesis that the hot-carrier transfer into unoccupied orbitals in CO₂ for C-O bond dissociation can be hardly occurred. Based on the above two experiments, we can concluded that CO production performance originates from the synergetic effect between H⁺ and hot e⁻.

To further understand the reaction mechanism, the effect of irradiation intensity on the thermal-photocatalytic performance of Bi-H_x was examined (**Fig. R10**). The strong dependence of the evolution rate of CO on the light intensity suggests that the photogenerated charges were responsible

for the CO₂ reduction. This result also mean that synchronous production of reactive H⁺ and e⁻ pairs in Bi-H_x was achieved by the consumption of photogenerated holes by the stored H.

Moreover, the XPS curves of Bi-H_x/NF after different thermo-photocatalytic reaction times were measured (Fig. R11). It can be seen that the XPS band ascribed to the oxidation state of Bi progressively blue-shifts with increasing reaction time, which is consistent with the fact that the consumption of H species proceeds during the CO₂ reduction reaction on Bi-H_x/NF. The above experiments, taken together, provide a solid evidence that the proton-assisted electron transfer approach indeed took place in the CO₂ reduction process of Bi-H_x/NF as shown in Fig. 5a-c, which is consistent with the DFT calculations in Fig. 6.

Fig. R8 CO production rates on the Bi-H_x/NF samples at 180 °C without light illumination by using H₂O (200 μL). (This figure has been added into the revised Supplementary Information as new Fig. S7.)

Fig. R9 CO production rates on the Bi/NF samples under the thermal-photocatalytic conditions without using proton sources. (This figure has been added into the revised Supplementary Information as new Fig. S8.)

Fig. R10 Effect of irradiation intensity on CO evolution rate over Bi-H_x/NF at 180 °C. (This figure has been added into the revised Supplementary Information as new Fig. S9.)

Fig. R11 XPS spectra of Bi-H_x/NF after different thermo-photocatalytic reaction times. (This figure has been added into the revised Supplementary Information as new Fig. S11.)

Action (II): Based on the reviewer's comments, the proton-assisted electron transfer mechanism on Bi-H_x has been investigated step by step, which provided directly experimental evidence on this reaction mechanism. A detailed discussion was added in the revised manuscript on **Page 18-19** as shown in the following.

In order to experimentally prove that the thermal-photocatalytic CO₂ reduction on Bi-H_x was performed by the proton-assisted electron transfer approach, the following experiments were designed. First, CO₂ reduction on Bi-H_x/NF was performed at 180 °C without light illumination by using H₂O (200 μL) as proton source. The negligible CO production activity (Supplementary Fig. 7)

demonstrates that photoinduced charges are the primary factor for CO₂ reduction on Bi-H_x/NF. Second, under the thermal-photocatalytic conditions, the formation of negligible CO on the bare Bi/NF without using proton sources (Supplementary Fig. 8) confirms that the CO₂-to-CO conversion can hardly occur only with hot-electron transfer into unoccupied orbitals of adsorbed CO₂. Based on the above two experiments, we can conclude that the thermal-photocatalytic CO production performance on Bi-H_x/NF should originate from the synergetic effect between H⁺ and hot e⁻, which is accordant with previous reports¹⁻³. The above experiments, taken together, provide a solid evidence that the proton-assisted electron transfer approach indeed took place in the CO₂ reduction process of Bi-H_x/NF as shown in Fig. 5a-c, which is consistent with the DFT calculations in Fig. 6.

To further understand the reaction mechanism, the effect of irradiation intensity on the thermal-photocatalytic CO₂ reduction activity of Bi-H_x/NF was examined (Supplementary Fig. 9). The strong dependence of the evolution rate of CO on the light intensity agrees well with the proposed mechanism that the synchronous production of reactive H⁺ and e⁻ pairs in Bi-H_x was achieved by the consumption of photogenerated holes by the stored H atoms. Moreover, the XPS curves of Bi-H_x/NF after different thermo-photocatalytic reaction time were measured (Supplementary Fig. 10). It can be seen that the XPS band ascribed to the oxidation state of Bi progressively blue-shifts with increasing reaction time, which is consistent with the fact that the consumption of H species proceeds during the CO₂ reduction reaction on Bi-H_x/NF. As a result of the depletion of H species after 27 h reaction, oxidation state of Bi in Bi-H_x/NF returned to the state that is similar with Bi/NF without H storage (Supplementary Fig. 10). The reaction time dependent XPS spectra also agree well with the proton-assisted electron transfer mechanism for CO₂ reduction on Bi-H_x/NF. In addition, based on the XPS analysis of Bi-H_x/NF, no obvious change on the ratio of Bi⁰ and Bi³⁺ is observed after 27 h reaction, indicating the good stability of the catalyst under thermo-photocatalytic reaction conditions.

14. Overall, I am not convinced this brings the quality and novelty needed for Nature Communications.

Response: In this revised manuscript, we considered all the comments of reviewers, and have tried our best to revise the manuscript accordingly. In addition to the novel two-step approach for CO₂ conversion as involved in our original manuscript, we have addressed the ambiguities related to proton-coupled electron transfer process by designing new experiments and have strengthened the cooperative contribution of proton and electron transfer in the two-step catalysis, which enhance the mechanistic aspect of our paper and provide theoretical and experimental insights on this two-step approach. Furthermore, we also take a first step by overcoming the challenges related to the cost, abundance, and hence the feasibility of implementing plasmonic systems for light-driven photocatalysis of important chemical transformations, which could bring novelty of our manuscript with clarity. To improve and strengthen the manuscript, we performed additional discussion about this important point in the revised manuscript. We truly appreciate the reviewer to keep in mind these important points while reevaluating this work for Nature Communications.

Response to Reviewer 2's Comments:

1. In the present work, Y. Li et al. study a hydrogen-stored Bi (Bi-H_x) system as an electron proton-transfer mediator for CO₂-to-CO conversion. They claimed that the stored hydrogen not only can react with CO₂ molecule to produce H₂O and CO but also it plays a role as a hole scavenger to promote the charge separation. The overall reaction is carried out under light at a certain temperature (180 °C), resulted in a high CO production yield of 283.3 μmol g⁻¹ h⁻¹. Using DFT calculations, they show that the energy barrier via a proton-assisted electron transfer pathway is much smaller on the Bi-H_x than the pristine Bi. However, they miss a deeper discussion on the material characterizations that are reflected in my comments below. This manuscript reports original results and a novel idea. Hence, I think this work can be published in Nature Communications after major revisions.

Response: We appreciate positive comments on our manuscript. Based on the comments below, we have improved the manuscript and it has further raised the quality of the manuscript.

2. The CO₂ reduction process is carried out under light at temperatures much higher than room temperature, which is used in the conventional photocatalytic CO₂ reduction. Therefore, I suggest the authors use the term “thermo-photocatalytic” (or “photo-thermocatalytic”, depending on the primary importance of a thermo- or photo-effect) in the title and other related parts of the manuscript.

Response: We thank the review for this suggestion.

Action: The term “thermo-photocatalytic” was used in the title and other related parts of the manuscript.

3. (I) The XPS spectrum of the Bi-H_x was not fitted well (Fig. 2c): the FWHM of the oxidation state Bi³⁺ is much larger than that of metallic Bi. The same FWHM will cause a smaller blue shift relative to the pristine Bi. If they need to use different FWHM, it should be addressed in the manuscript. (II) Besides, the hydrogenation process resulted in a larger FWHM. Can it possibly suggest the presence of other oxidation states on the surface? (III) The authors should add deconvoluted carbon and oxygen spectra in the supporting information. These spectra can provide useful information about the probable existence of any additional oxidation states. (IV) As mentioned by authors (page 6 line 108-109), there are Bi₂O₃ at the very surface of the Bi-H_x. So, is it a kind of Bi-H_x@Bi₂O₃ core-shell system? How thick is this Bi₂O₃ “shell”?

Response: (I) The XPS measurements on the Bi-H_x and Bi were repeated. As shown in Fig. R12 below, the peaks belong to the Bi³⁺ and Bi⁰ can be clearly separated. The XPS data in Fig. R12 were similar with the result reported in the literature (*ACS Catal.* 2020, 10, 743–750), ensuring the correctness of this test. Furthermore, from the repeated XPS spectra in Fig. R11 above, it can be confirmed that similar spectra were obtained in XPS measurements of these samples, also confirming the correctness of the XPS tests in this time.

Response: (II) This problem was not existed in the new XPS data as shown in Fig. R12 below.

Response: (III) According to the suggestion of the reviewer, the deconvoluted oxygen spectra of the two samples are shown in Fig. R13 below. O 1s XPS spectra were deconvoluted into three peaks that are ascribed to Bi-O bands, surface hydroxyl oxygen, and adsorbed O₂, which further confirms the generation of Bi-O in Bi₂O₃ on the surface of the two samples.

In addition, the C 1s XPS spectra of the two samples were also examined as shown in **Fig. R14** below. The two samples show similar C 1s XPS spectra which is consistent with the XPS results of Bi and O. The carbon species on the two sample can deconvoluted into three peaks corresponding to C–C, C–O, and C=O bonds, respectively (*Chem. Eng. J.* 2019, 374, 231), which is mainly caused by the adsorbed CO₂. Based on this fact, the C 1s XPS spectra might be not helpful in understanding the oxidation state of Bi. Therefore, the C 1s XPS spectra were not included in the revised manuscript.

Response: (IV) To investigate the thickness of Bi₂O₃ on Bi, the transmission electron microscope (TEM) analysis was performed. As shown in **Fig. R15** below, amorphous Bi₂O₃ layers with a thickness of ~3 nm were clearly seen on the out surface of Bi of Bi-H_x and Bi, respectively.

Fig. R12 High-resolution Bi 4f XPS spectra of Bi/NF and Bi-H_x/NF. (This figure has been updated in the revised manuscript as Fig. 2d.)

Fig. R13 High-resolution XPS spectra of O 1s of Bi/NF and Bi-H_x/NF. (This figure has been added into the revised Supplementary Information as new Fig. S1.)

Fig. R14 High-resolution XPS spectra of C 1s of Bi/NF and Bi-H_x/NF.

Fig. R15 HRTEM images taken from the edge of the **a** Bi and **b** Bi-H_x nanosheets. (This figure has been added into the revised manuscript as new Fig. 3h and i.)

Action: As shown in **Fig. R12**, an updated figure on XPS measurements has been embedded in the revised manuscript. The high-resolution spectrum of oxygen spectra of Bi-H_x and Bi were added in the supporting information (**Supplementary Fig. 1**), which indicate that only Bi₂O₃ was formed on the surface of Bi-H_x and Bi. The HRETEM image at the edge of Bi-H_x and Bi were examined, and Bi₂O₃ layers with a thickness of ~3 nm were clearly seen as shown in **Fig. R15**. Moreover, the corresponding discussions were added in the revised manuscript on **Pages 7 and 8**.

High-resolution XPS spectra of O 1s for the two samples were examined to further confirm the surface chemistry of Bi (Supplementary Fig. 1). The two samples exhibit similar O 1s XPS spectra that can be deconvoluted into three peaks corresponding to, Bi-O bands (529.3 eV), surface hydroxyl

oxygen (530.8 eV), and adsorbed O₂ (532.7 eV), further confirming for the generation of Bi–O in Bi₂O₃ on the surface of the two samples²⁴.

As shown by the XPS spectra in Fig. 2d, the surface of Bi can be easily oxidized into Bi₂O₃ in the air. To investigate the formed Bi₂O₃ layer, the TEM analysis was performed on the edge of the Bi-H_x and Bi sheets. As shown in Fig. 3h and i, amorphous Bi₂O₃ layers with a thickness of ~3 nm were clearly formed on both surfaces of Bi-H_x and Bi.

4. Bi₂O₃ has a bandgap of about 2.5-2.7 eV. Therefore, it can show a peak at around 450-500 nm. So, it seems that the broad peak between 400-600 (in Fig. 2d) is due to the convolution of both LSPR of Bi-H_x and partial Bi₂O₃ absorption. If the nature of the peak is LSPR, the authors can provide several samples with different particle sizes (by using different electrodeposition times from 3 to 15 min). Then, if the nature of the absorption is LSPR, they would observe a shift in the UV-visible absorption.

Response: According to the suggestion of the reviewer, the morphology of Bi-H_x nanostructures were tuned by varying the deposition time (1, 4, and 15 min). SEM images in Fig. R16 below show that the size of the Bi sample was great affected by the deposition time. XRD measurements show that elemental Bi was formed at these situations as shown in Fig. R17a below. The optical absorptions of the obtained samples in Fig. R17b below shows that the localized surface plasmon resonance (LSPR) effect of a nanostructure is strongly correlated to the size of the sample. For example, no LSPR absorption was shown on the nanoparticulate of Bi with a diameter of about 100 nm, and the LSPR peak increases in intensity and red-shifts with increasing size of Bi. This phenomenon confirms that the absorption peak in Fig. 3h in revised manuscript is indeed caused by the LSPR absorption of Bi.

Fig. R16 SEM images of Bi/NF prepared under deposition time of (a) 1 and (b) 4 min. (This figure has been added into the revised Supplementary Information as new Fig. S2)

Fig. R17 (a) XRD patterns and (b) UV-visible adsorption spectra of Bi/NF samples prepared at different deposition time. (This figure has been added into the revised Supplementary Information as new Fig. S3.)

Action: In the revised manuscript, the Bi nanostructures with different size were synthesized by varying deposition time. It is proved that the LSPR effect of a nanostructure is strongly correlated to the size of the sample. Furthermore, we have provided a detailed discussion on the size depended LSPR in the revised manuscript **on Page 8-9**. For additional coherency, we have moved Fig. 2d in the original version into Fig.3h in the revised manuscript.

To prove that this peak is caused by the localized surface plasmon resonance (LSPR) absorptions, the Bi nanostructures with sizes of ~ 100 and ~ 400 nm were further synthesized at the deposition time of 1 and 4 min, respectively (Supplementary Fig. 2). XRD measurements show that elemental Bi was formed at these situations (Supplementary Fig. 3a). Then, the corresponding absorption spectra of the obtained Bi/NF at different deposition time were tested (Supplementary Fig. 3b). The Bi sample with a diameter of ~ 100 nm shows a bandgap absorption onset of ~ 600 nm, whereas the absorption peak at approximately 480 nm in Fig. 3h disappears. This optical phenomenon implies that the absorption peak between 400 and 600 nm in Fig. 3h was induced by LSPR, rather than from the coupling effect of Bi and Bi_2O_3 layer. Moreover, by comparing the spectra of Bi samples with sizes of ~ 100 nm and 400 nm, we can find that the position of the absorption peak shows a red-shift with the size increasing (Supplementary Fig. 3b), which is consistent with the characteristic of LSPR absorption that is strongly correlated to the shape and size of the materials²⁵. The above optical properties of the samples confirm that the absorption peak at 480 nm in Fig. 3h can be attributed to the LSPR of Bi, which is consistent with the previous report²⁶.

5. SAED patterns should be recorded at the edge and basal plane of both Bi-H_x and Bi particles. It can show if there are any other local secondary phases or not.

Response: As suggested, the structures of Bi-H_x and Bi particles were characterized by selected-area electron diffraction as shown in **Fig. R18** below, which confirmed the single crystalline nature of the samples. Based on the results in SAED and HRTEM, the as-synthesized products consist of a single Bi phase except the amorphous Bi₂O₃ layers.

Fig. R18 SAED patterns of **a** Bi/NF and **b** Bi-H_x/NF. (This figure has been added into the revised manuscript as new Fig. 3h and i.)

Action: We have now included the corresponding SAED patterns in Fig. 3 in the manuscript. The corresponding discussions were added in the revised manuscript on **Page 8**.

The SAED patterns of Bi (insert of Fig. 3h) and Bi-H_x (insert of Fig. 3i) show that only the spots ascribing to the rhombohedral Bi are observed, suggesting the single crystalline nature of the samples.

6. The higher activity of the Bi-H_x/NF could be due to the formation of nanopores during the hydrogenation process, rather than, or not only just because of the proton-assisted electron transfer effect as claimed by the authors. Therefore, the authors should provide direct experimental evidence that the overall surface areas of both samples (Bi/NF and Bi-H_x/NF) are almost the same.

Response: We have measured the specific surface areas of Bi₂O_{3-x} and Bi. The result showed that the specific surface areas of Bi-H_x and Bi were 3.3356 m²·g⁻¹ and 2.1039 m²·g⁻¹, respectively, demonstrating that the improved CO₂ reduction activity of Bi-H_x/NF cannot be mainly ascribed to the effect of surface area.

Action: The corresponding results are added in the revised manuscript on **Page 14** as:

In addition, the specific surface areas of the two samples were studied by the Brunauer-Emmett-Teller (BET) method based on the nitrogen adsorption isotherm, showing that the BET surface areas of Bi-H_x and Bi are 3.3356 m²•g⁻¹ and 2.1039 m²•g⁻¹, respectively. This result demonstrates that the surface area should not be the main factor for the enhanced CO₂ reduction activity of Bi-H_x/NF.

7. (I) For CO₂ reduction, the authors had applied heat, hence, there is additional input energy (ΔE), which should be added to the denominator of the AQE. In other words, the authors report the AQE values only based on photo-, but neglecting the thermo- contribution completely (Fig. 4f). (II) I will also suggest the authors to do the test under AM 1.5 solar light (only photo-catalysis, but not thermo-photo-catalysis). How much will the AQE be for photo-catalysis? The result shall be compared to other reported systems.

Response (I): According to the reviewer's comments, thermal contribution considered in the calculation of AQEs. The radiative power of heat per unit area was given by the Stefan–Boltzmann law: $Q = \sigma(T_r^4 - T_0^4)$, where T_r is the reaction temperature in K, and σ is Stefan–Boltzmann constant ($5.67 \times 10^{-8} \text{ W}\cdot\text{m}^{-2}\cdot\text{K}^{-4}$). The maximal work done by the thermal energy was calculated by using the Carnot equation, $W = Q (1 - T_0/T_r)$, where T_0 is the ambient temperature for performing the CO₂ reduction (299.15 K). Based on the above two equations, the real contribution of heat in CO₂ reduction was calculated to be 65.8 mW cm⁻².

Action (I): According to this new calculation method, the AQEs in Fig. 4f were revised. We have provided the details about this calculation method in the Methods of the revised manuscript, which provides reference for calculating AQEs of thermo-photocatalytic reaction in the future. The corresponding description on this method was added in the revised manuscript on **Page 26**.

Thermal contribution at 180 °C was considered in the calculation of AQEs. The radiative power of heat per unit area can be described by the Stefan–Boltzmann law⁵³,

$$Q = \sigma(T_r^4 - T_0^4) \quad (8)$$

where T_r is the reaction temperature in K, and σ is Stefan–Boltzmann constant ($5.67 \times 10^{-8} \text{ W}\cdot\text{m}^{-2}\cdot\text{K}^{-4}$). The maximal work done by the thermal energy was calculated by using the Carnot equation⁵⁴,

$$W_h = Q (1 - T_0/T_r) \quad (9)$$

where T_0 is the ambient temperature for performing the CO₂ reduction (299.15 K). Based on the above two equations, the real contribution of heat at 180 °C in CO₂ reduction was calculated to be 65.8 mW•cm⁻². The AQE was calculated by the following equation:

$$\begin{aligned}
\text{AQE}(\%) &= \frac{\text{Number of reacted electrons}}{\text{Number of incident photons}} \times 100\% \\
&= \frac{\text{Number of evolved CO molecules} \times 2}{\text{Number of incident photons}} \times 100\% \\
&= \frac{M \times N_A \times 2}{\frac{W \times A \times t \times \lambda}{h \times c}} \times 100\% \\
&= \frac{M \times N_A \times 2}{\frac{(W_l + W_h) \times A \times t \times \lambda}{h \times c}} \times 100\%
\end{aligned} \tag{10}$$

where M is the CO evolution rate (mol/s), N_A is Avogadro constant, W is the total energy input (W), A is irradiation area (m^2), t is the time of light illumination (s), λ is the corresponding wavelength (m), h is $6.62 \times 10^{-34} \text{ J}\cdot\text{s}^{-1}$, and c is $3.0 \times 10^8 \text{ m}\cdot\text{s}^{-1}$.

Response (II): According to the suggestion, photocatalytic CO_2 reduction on Bi- H_x /NF was carried out under AM 1.5 solar light irradiation (100 mW cm^{-2}) at ambient temperature. However, negligible amount of CO was produced as shown in **Fig. R19**. This result is not beyond expectation because Bi is a plasmonic metal photocatalysts. To the best our knowledge, all the reported plasmonic photocatalysts based on metal in different reactions, such as CO_2 reduction, can only work when the reaction temperature is relatively high ($>150 \text{ }^\circ\text{C}$). For example, the reaction temperatures for photocatalytic CO_2 conversion are $200\text{-}350 \text{ }^\circ\text{C}$ for plasmonic Au, and Rh (*Nat. Commun.* 2017, 8, 14542), $400 \text{ }^\circ\text{C}$ for plasmonic Pt-Au (*ACS Appl. Mater. Interfaces* 2018, 10, 408).

Fig. R19 CO production on Bi- H_x /NF under AM 1.5 solar light irradiation.

Action (II): Because of the negligible activity of sample for CO_2 reduction under AM 1.5 solar light irradiation, the result was not included in the revised manuscript.

8. Current data did not provide convincing evidence that the hydrogen (in Bi-H_x lattice) can have bi-functional roles (page 12 line 231). Especially, for the claimed point 2, the authors should do the XPS after GC test for different times to prove that there is a continuous shift that can probably be assigned to the consumption of H (i.e. H element belonging to the Bi-H_x phase).

Response: We thank the review for this suggestion, which is helpful for understanding the CO₂ reduction mechanism on Bi-H_x. According to the suggestion, the XPS curves of Bi-H_x at different reaction times were tested. As shown in Fig. R20 below, the XPS band progressively blue-shifts with increasing reaction time, which is clearly attributable to a losing of H species.

Fig. R20 XPS spectra of Bi-H_x at different thermo-photocatalytic reaction time. (This figure has been added into the revised Supplementary Information as new Fig. S10.)

Action: To incorporate the new experimental results, the following discussions were added into the revised manuscript on **Pages 19** as:

Moreover, the XPS curves of Bi-H_x/NF after different thermo-photocatalytic reaction time were measured (Supplementary Fig. 10). It can be seen that the XPS band ascribed to the oxidation state of Bi progressively blue-shifts with increasing reaction time, which is consistent with the fact that the consumption of H species proceeds during the CO₂ reduction reaction on Bi-H_x/NF. As a result of the depletion of H species after 27 h reaction, oxidation state of Bi in Bi-H_x/NF returned to the state that is similar with Bi/NF without H storage (Supplementary Fig. 10). The reaction time dependent XPS spectra also agree well with the proton-assisted electron transfer mechanism for CO₂ reduction on Bi-H_x/NF. In addition, based on the XPS analysis of Bi-H_x/NF, no obvious change on the ratio of Bi⁰ and Bi³⁺ is observed after 27 h reaction, indicating the good stability of the catalyst under thermo-photocatalytic reaction conditions.

9. The PEC measurements were not carried out in a stable condition. Degradation of the maximum photocurrent density is about 40% after several minutes (Fig. 5d)! The authors need to explain why? Or repeat the experiment.

Response: According to the suggestion, the experiment was repeated accordingly. As shown in **Fig. R22** below, no obvious degradation on photocurrent density was observed in this time.

Fig. R21 Photocurrent transient responses of Bi/NF and Bi-H_x/NF catalysts under 420 nm LED light irradiation at a bias of 1.07 V (vs. RHE). (This figure has been updated in the revised manuscript as Fig. 5d.)

Action: We accordingly improved the quality of graphics of Fig. 5d in the revised manuscript as shown in Fig. R21.

10. As shown in XPS spectra, the very surface contains the Bi₂O₃ phase (quite dominant Bi³⁺ actually), which can change the calculated free energies (Fig. 6b). A serious concern is raised for the potential impact of this surface oxide layer on the whole CO₂ reduction process when it is in the vicinity of the Bi-H_x phase. The authors have claimed the energy profiles are not dramatically affected by the presence of Bi oxide (comparing Supplementary Fig. 2b and Main Fig. 6b), however, the computational results can depend on the content of oxide in the model. I think the authors must provide clear evidence, preferably experimental, showing that the surface oxide layer has not had a big impact on the proposed reaction pathway.

Response: In order to experimentally prove that the surface oxide layer have a negligible effect on the CO₂ reduction process, the Bi(0) on the surface Bi-H_x sample was transferred into Bi₂O₃ by treating the sample in air at 80 °C for 0.5 h. As proved by the XPS in **Fig. R22** below, only the peaks belonging to Bi(3) were observed, indicating the surface of the sample was fully covered by Bi₂O₃ layer after the heat treatment. Then, the controlled experiment for CO₂ reduction was carried out on the Bi-H_x sample after heat treatment. By comparing the activity in **Fig. R23** below with that in Fig 4d in the revised manuscript, no catalytic performance degradation was found when Bi(0) was replaced by Bi(3), indicating that the formed Bi₂O₃ layer in air did not affect the CO₂ reduction activity on Bi-H_x/NF, which is consistent with the DFT calculations in Fig. 6. Despite the ~3 nm amorphous Bi₂O₃ layer surrounding the Bi core, tunneling of the hot carrier to the oxide surface is still viable due

to a high-density of defect states in the amorphous Bi_2O_3 layer and a high energy of the hot charge carriers produced by localized surface plasmon resonance (*Adv. Mater.* 2020, 2000086).

Fig. R22 High-resolution XPS spectra of Bi 4f of $\text{Bi-H}_x/\text{NF}$ sample after heat treatment. (This figure has been added into the revised Supplementary Information as new Fig. S6.)

Fig. R23 CO evolution curve on the $\text{Bi-H}_x/\text{NF}$ catalyst treated in air at $80\text{ }^\circ\text{C}$ for 0.5 h. (This figure has been added into the revised manuscript as new Fig. 4f.)

Action: The photocatalytic CO_2 reduction activity of the $\text{Bi-H}_x/\text{NF}$ sample heated in air at $80\text{ }^\circ\text{C}$ for 0.5 h was tested and shown in **Fig. R23**. We have provided corresponding detailed discussion in the revised manuscript on **Pages 13-14** as:

As shown in the XPS spectra in Fig. 2d, the amount of Bi^0 species on the surface of $\text{Bi-H}_x/\text{NF}$ is much larger than that of Bi/NF . In order to exclude that the enhanced activity of $\text{Bi-H}_x/\text{NF}$ in Fig 4e is not caused by the increased content of Bi^0 , the Bi^0 on the surface of $\text{Bi-H}_x/\text{NF}$ was oxidized into Bi^{3+} by treating the sample in air at $80\text{ }^\circ\text{C}$ for 0.5 h. The absence of Bi^0 species on the surface of $\text{Bi-H}_x/\text{NF}$ after the treatment can be proved by XPS measurement (Supplementary Fig. 6). In the following, the

controlled experiment for CO₂ reduction was carried out on the sample after heat treatment. Compared with the activity of Bi-H_x/NF in Fig. 4d, no performance degradation on CO production was found on Bi-H_x/NF when the surface Bi⁰ species was replaced by Bi³⁺ (Fig. 4f), indicating that the improved activity of Bi-H_x/NF in Fig. 4e cannot be attributed to the increased amount of surface Bi⁰. Based on the above results, it can also speculate that the automatically formed Bi₂O₃ layer on Bi-H_x/NF in air plays negligible effect on the CO₂ reduction activity, which is consistent with the DFT calculations in Fig. 6. For the Bi-H_x covered by ~3 nm amorphous Bi₂O₃ layer, tunneling of the hot carrier to the out surface is still feasible due to a high energy of the hot charge carriers generated by LSPR and a high-density of defect states in the amorphous layer. A similar result has been observed for Bi catalysts for electrochemical H₂ production and CO₂ reduction in previous reports^{22,24}.

11. What will happen if the authors do the same experiment in CO₂ and H₂O atmosphere? Can the introduction of H₂O keep the concentration of hydrogen constant in the Bi-H_x system and provide better stability?

Response: According to the reviewer's comments, we performed the CO₂ reduction reaction in CO₂ and H₂O atmosphere by adding 200 μL H₂O in the reactor before the thermo-photocatalytic reaction. As shown in Fig. R24 below, the CO₂ production activity stability of Bi-H_x/NF in CO₂ + H₂O + Ar atmosphere is similar with that in CO₂ + Ar. This result indicated that the introduction of H₂O cannot keep the concentration of hydrogen constant in the Bi-H_x system. This is easily understand based on the fact that the state of H in H₂O is different from that in Bi-H_x/NF. This is a good idea that need to be systematically studied in our future work.

Fig. R24 CO production on Bi-H_x/NF in CO₂ + H₂O + Ar atmosphere.

Action: Considering that the introduction of H₂O induced negligible effect on the CO₂ reduction performance of Bi-H_x/NF, this result was not included in the revised manuscript.

Response to Reviewer 3's Comments:

Dear Authors

Find here my concerns.

Response: Thanks very much for your feedback. We have addressed fully the comments to improve the quality of our manuscript.

1. First and foremost for any photocatalytic reaction needs to analyze the band gap of material which is not provided in manuscript.

Response: The present Bi nanosheets are a special metal catalyst that has the surface plasmon resonance (SPR) property. For metal nanostructures with high free electron mobility, there is an inherent oscillation frequency of valence electrons against the restoring force of the positively charged nuclei. SPR is excited when the frequency of incident photon matches the natural oscillation frequency of valence electrons, leading to the coherent oscillation of electrons in energy and space. From above discussion, we can conclude that SPR is a unique optical property based on the free valence electrons that is different from traditional semiconductor based on the excitation from band gap. Therefore, the band gap of Bi was not provided in manuscript.

2. In Fig 2d UV-Vis analysis doesn't show any significant difference between both catalyst then how both can behave differently prove it by providing additional analysis data like PL (Photo luminescence analysis) that can show the recombination of excitons.

Response: Based on reviewer's suggestion, photoluminescence (PL) spectra of Bi-H_x/NF and Bi/NF were obtained on a Fluorescence spectrophotometer (F-7000, Hitachi, Japan) at room temperature. Steady-state PL spectra of the as-prepared samples were shown in **Fig. R25** below. It can be seen that the PL intensity of Bi-H_x/NF was significantly lower than that of Bi/NF, indicating a much lower recombination rate of charge carriers and a faster charge transfer within the sample Bi-H_x/NF, relative to sample Bi/NF. This result strongly suggested that the charge separation efficiency of Bi/NF can be improved by the hydrogen storage process.

Fig. R25 Steady-state photoluminescence (PL) spectra of Bi/NF and Bi-H_x/NF. (This figure has been added into the revised manuscript as new Fig. 5e.)

Action: To address this question and incorporate the new results, the following sentences were added into the revised manuscript on **Page 18** as:

To further study the separation of photoinduced electron-hole pairs under light illumination, steady-state photoluminescence (PL) analysis was performed on the two samples. As shown in Fig. 5e, the PL intensity of Bi-H_x/NF is significantly decreased in compared with that of Bi/NF, indicating that the incorporation of H atoms on Bi/NF has effectively suppressed the radiation recombination of charge carriers, which is helpful for CO₂ reduction reaction on Bi-H_x/NF.

3. From Fig 2b cyclic voltametric analysis nickel foam is not much sensitive toward electrocatalytic reaction thus the use of Nickel foam must be explained.

Response: Nickel foam, which has the advantages of high porosity, superior performance of fluid penetration, and good mechanical performance, is a suitable substrate for depositing Bi catalyst.

Action: To address this question, the following sentences were added into the revised manuscript on **Page 5** as:

The NF was chosen as substrate for depositing Bi because of its good mechanical performance and high porosity with interconnected framework structure that is favourable for easy contact between electrolyte and electrode, and fast ion transport.

4. From Fig 2b, it can also clearly observed the absorption/desorption peak of hydrogen that also questioned the stability of Bi-H_x/NF catalyst.

Response: We thank the reviewer for this comment. The cyclic voltammetry (CV) curves in Fig. 2b were shown to prove the electrochemical hydrogen storage behavior of Bi/NF. The CV curve in Fig. 2b is from Bi/NF and not from Bi-H_x/NF. In the present work, the Bi/NF catalyst was used for CO₂ reduction by reversible storage and release of hydrogen. Therefore, the stored H in Bi-H_x/NF should be easily released. In this respect, an appropriate stability of Bi-H_x/NF is needed. In addition, stability of Bi/NF in the overall CO₂ reduction process has been fully proved in Fig. 4d, in which no obvious decrease in CO₂ reduction activity was observed after 81 h of reaction.

5. Fig. 2c XPS analysis shows the significant shift of binding energy though other analysis though there is no extra oxidation state of Bi is reported thus the change must be explained.

Response: The significant shift of binding energy should be due to the electron donation of hydrogen atoms.

Action: To address this question, the following sentences were added into the revised manuscript on **Page 7** as:

However, for Bi-H_x/NF, the peaks belonging to the oxidation states of Bi are obviously red-shifted relative to those for Bi/NF, which is related to incomplete oxidation of Bi due to the electron donation of hydrogen atoms.

6. GC analysis presented the production of CO and hydrogen molecule in the system while photocatalytic reduction does not selective that clearly give the information about formation of other products or hydrocarbons, it require some additional analysis.

Response: The result on GC-MS also indicated that except for CO, no other products, such as CH₄ and CH₃OH, were detected as shown in Fig. 4c.

Action: To address this question, formation of other hydrocarbons was evaluated by GC, and GC-MS. The following sentences for describing the result and the detailed analysis method used were added into the revised manuscript on **Page 10, 12, and 25**.

Other possible products (such as CH₄, and CH₃OH) were not detected, confirming that high selectivity for CO₂ reduction was achieved on Bi-H_x/NF.

As shown in Fig. 4c, except for the peaks belonging to CO₂ and CO, no other peaks were observed in Fig. 4c, further confirming the high selectivity for CO₂ reduction on Bi-H_x/NF. This result is consistent with that of gas chromatography.

For detecting CH₃OH, the gaseous products from the reactor were analysed by an offline GC (Fuli Corp., China) equipped with a flame ionized detector (FID). The equipped column was TDX-01.

7. Comparative analysis data of ¹³CO and CO is not provided in manuscript to prove the reliability of reaction mechanism needs to be rechecked.

Response: For comparison, the isotopic ¹²CO₂ labelling experiment was carried out on Bi-H_x/NF in the revised manuscript. As shown in **Fig. R26** below, ¹²CO with m/z = 28 is the main product for ¹²CO₂ reduction, which further confirms that CO was produced from the thermo-photocatalytic reduction of CO₂ on Bi-H_x/NF.

Fig. R26 Mass spectrum from GC-MS analysis of the CO generated in the catalytic CO₂ reduction reaction using ¹²CO₂. (This figure has been added into the revised Supplementary Information as new Fig. S4 .)

Action: To address this question, the following sentences were added into the revised manuscript on **Page 12** as:

For comparison, the isotopic $^{12}\text{CO}_2$ labelling experiment was also performed on Bi- H_x /NF (Supplementary Fig. 4), and ^{12}CO with $m/z = 28$ is found to be the main product for $^{12}\text{CO}_2$ reduction, which further demonstrates that CO was produced from the thermo-photocatalytic reduction of CO_2 on Bi- H_x /NF.

8. Dissolution of gas in the water requires some extra outer forces to maintain an equilibrium that is not explained in the manuscript need to be recheck and optimize the reactor conditions.

Response: We thank the reviewer for this comment. This might be caused by our unclear description on the overall CO_2 reaction process. The overall CO_2 reaction process contains two steps. In the first step, splitting of H_2O into H atoms and O_2 and storage of the H atoms in Bi nanoparticles (denoted as Bi- H_x) were simultaneously achieved by an electrochemical approach. In the second step, the obtained Bi- H_x was used as a reducing equivalent to reduce CO_2 to CO by *in situ* generating H^+/e^- pairs under light irradiation. In fact, the two step reactions were proceeded in two separate reactors as described in the Methods part of the manuscript. The first reaction for electrochemical H storage on Bi was performed in an aqueous solution of KOH (1 M) electrolyte. After that, the formed Bi- H_x /NF was used for CO_2 reduction in another gas-solid phase reactor. Therefore, no water was present in the CO_2 reduction process after the formation of Bi- H_x .

Action: For clarity, the unclear sentences were changed in the revised manuscript on **Page 10 and 11** as:

The sentence “The CO_2 reduction performance of Bi- H_x /NF was evaluated in a 300 mL closed chamber with a quartz window on top, in which both the temperature and light illumination could be controlled.” in the original manuscript on Page 7 was changed to:

After the electrochemical hydrogen storage process of Bi/NF in 1 M KOH electrolyte, the formed Bi- H_x /NF was used for CO_2 reduction in another gas-solid reactor with a quartz window on top, in which both the temperature and light illumination could be controlled.

The sentence “As a control experiment, pristine NF was treated at a reduction potential of -1.2 V for 10 min in 1 M KOH electrolyte and placed in the same reactor for testing the CO_2 reduction activity.” in the original manuscript on Page 8 was changed to:

As a control experiment, pristine NF was treated at a reduction potential of -0.18 V for 10 min in 1 M KOH electrolyte and then placed in the gas-solid reactor for testing the thermo-photocatalytic CO_2 reduction activity.

9. In DFT analysis favors the mechanism of reaction but the same time free energy value with respect to change in the transition, state shows reaction happened under the artificial condition, it was not natural that need to be explained.

Response: Sorry for the misunderstanding. In our manuscript, the free energy value is not with respect to the change in the transition state. In fact, free energy and transition state are two independent factors to determine the reaction activity. Generally, free energy diagram does not include the transition state, and only based on the thermodynamic properties of the reaction intermediates. According to the free energy diagram the thermodynamic barrier can be predicted. Furthermore, by including entropy correction, the temperature effect can also be considered based on the experimental conditions (here, 298 K is chosen). Therefore, it is not under the artificial condition.

Action: In the revised version we emphasize and distinguish the roles of free energy and transition state.

In the section of computational details,

The sentence “The transition state (TS) structures and the reaction pathways were located using the climbing image nudged elastic band (CI-NEB) method⁴⁵.” is changed to “*To explain the reaction activity from kinetic aspect, the transition state (TS) structures and the reaction pathways were located using the climbing image nudged elastic band (CI-NEB) method⁵⁹.*”

The sentence “To illustrate the CO₂ reduction reaction activity, the free energy diagrams were estimated...” is changed to “*To illustrate the CO₂ reduction reaction activity from thermodynamic aspect, the free energy diagrams were estimated...*”

In addition, in the discussion section, we replace “barrier” calculated from transition state and free energy diagram to “*kinetic barrier*” and “*thermodynamic barrier*”, respectively.

The sentence “Furthermore, according to the calculation results, H₂ dissociation is an endothermic process with a barrier of 1.09 eV (Fig. 6a)” is changed to “*Furthermore, according to the calculation results, H₂ dissociation is an endothermic process with an energy of 1.09 eV (Fig. 6a)*”

The sentence “Compared with the photoreduction of CO₂ on Bi in H₂ gas, the CO₂ conversion on Bi-H_x avoids the H₂ dissociation step, which might lower the energy barrier for CO₂ reduction.” is changed to “*Compared with the photoreduction of CO₂ on Bi in H₂ gas, the CO₂ conversion on Bi-H_x avoids the unfavourable H₂ dissociation step and the stored H might directly react with CO₂.*”

The sentence “Furthermore, the conversion of CO₂ to CO on Bi-H_x/NF via the COOH* intermediate even gives a much smaller thermodynamic barrier than H₂ decomposition on Bi/NF (1.09 eV), as shown in Fig. 6a.” is deleted to avoid misunderstanding.

REVIEWERS' COMMENTS

Reviewer #3 (Remarks to the Author):

Dear Editor

I don't have any concern over the response and revision. I felt the manuscript could be accepted for publication.